# IS YOUR LLM REALLY MASTERING THE CONCEPT? A MULTI-AGENT BENCHMARK

## ABSTRACT

Concepts are generalized abstractions that allow humans to categorize and reason efficiently. Whether Large Language Models (LLMs) possess a similar understanding of conceptual relationships, however, is not yet well established. Existing benchmarks primarily focus on factual recall or narrow tasks (*e.g.*, multiple-choice question answering or knowledge quizzes), offering limited insight into whether models understand conceptual relationships and subtle distinctions (*e.g.*, poetry *vs.* prose). Many also rely on static datasets that risk overfitting. To address this gap, we introduce CK-Arena, a multi-agent interaction benchmark inspired by the Undercover game, designed to evaluate the mastery of conceptual feature knowledge by LLMs. In CK-Arena, models must describe, differentiate, and infer distinguishing features of concepts from partial information, testing their ability to reason about both commonalities and differences across concept boundaries. The benchmark offers scalable datasets, rigorous evaluation protocols, and flexible extension methods, enabling comprehensive assessment of LLMs' conceptual understanding across multiple dimensions. Experimental results show that LLMs' understanding of conceptual knowledge varies significantly across different categories and is not strictly aligned with general model capabilities. The code is made publicly available at: `https://anonymous.4open.science/r/CK-Arena/readme.md`.

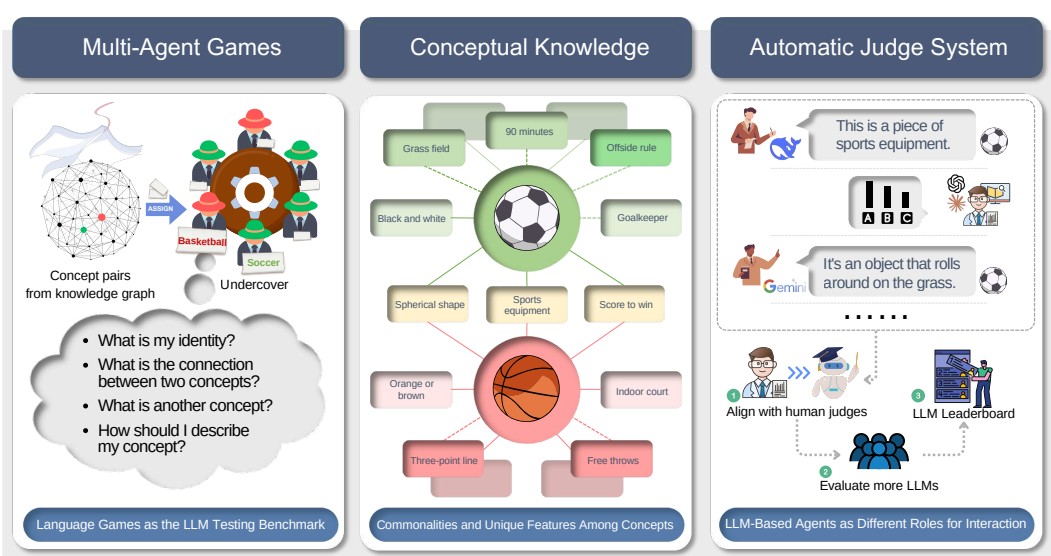

Figure 1: **Conceptual knowledge arena (CK-Arena).** The benchmark aims to evaluate the mastery of conceptual feature knowledge by LLMs. It builds on the interactive game *Undercover*, where concept pairs with overlapping and distinct features are assigned to LLM agents. Acting as players, models generate descriptions, infer similarities and differences, and make strategic decisions with partial information. As judges, they evaluate responses for sentence metrics. This multi-agent setup creates a dynamic and scalable environment for assessing conceptual understanding.

# 1 INTRODUCTION

A concept is a high-level abstraction of knowledge that captures shared properties of entities and their characteristic attributes. Understanding concepts requires recognizing their relationships as well as the similarities and differences that distinguish closely related ones, which is a fundamental aspect of human cognition (Wu et al., 2012; Gong et al., 2016; Ji et al., 2019; Zhang et al., 2021; Wang et al., 2024; Cao et al., 2024). For example, the concept *Primates* unites animals such as *monkeys* and *apes* through features like opposable thumbs, forward-facing eyes, and advanced cognitive abilities, while also involving subtle distinctions such as the presence of tails in most *monkeys* but not in *apes*. Human cognition naturally uses such conceptual structures for reasoning and adaptation, but it is still uncertain to what extent Large Language Models (LLMs) internalize and exploit these abstractions.

Recent work has highlighted the importance of conceptual knowledge as a core aspect of intelligence. Studies have examined conceptual design generation (Ma et al., 2023), concept editing (Wang et al., 2024), and abstract concept understanding (Liao et al., 2023; Chen et al., 2025), showing growing interest in concept-based reasoning for LLMs. Yet progress remains constrained by the lack of systematic benchmarks. Traditional benchmarks have advanced LLMs performance (Hendrycks et al.; Zellers et al., 2019; Liang et al.; Mostafazadeh et al., 2017), but most rely on static question–answer formats that test token-level accuracy and factual recall. These evaluations reduce knowledge to isolated items and mainly measure information retrieval, offering little evidence of whether models understand conceptual relationships or can distinguish closely related concepts. For example, a model may identify that *monkeys* and *apes* both belong to *Primates*, but this does not show understanding of the hierarchical relationships or distinctive features between the two groups. In addition, fixed formats such as multiple-choice questions provide only a partial view of reasoning, and the reliance on static datasets limits scalability, since building and updating them requires extensive human annotation.

Interactive game-based environments have emerged as an alternative (Lin et al., 2023; Zhou et al., 2023; Wu et al., 2023), offering dynamic contexts for multi-step reasoning. However, most existing simulations emphasize strategy, providing limited insight into whether models can represent and communicate conceptual knowledge. These gaps call for a systematic and scalable benchmark that directly evaluates conceptual reasoning in realistic interactive settings. To address this gap, we introduce CK-Arena, an interactive multi-agent benchmark for evaluating LLMs' ability to represent, differentiate, and communicate conceptual knowledge (Figure 1). We evaluate the LLMs by having them play the *undercover (Who is the spy?)* game, a multi-agent language game that involves describing a targeting word and identifying each player's role, and by assessing their multi-turn performance as well as their in-game statements. Unlike traditional dataset-based or strategy-focused benchmarks, CK-Arena engages models with concept pairs that share both overlapping and distinctive features, and offers scalable datasets, systematic evaluation protocols, and extensible tools for assessing conceptual understanding.

For evaluation, LLMs serve as referees and are combined with human calibration to ensure reliability. We test a set of recent language models over multiple rounds using a convergent rating system, producing an intuitive leaderboard of their relative performance. Beyond overall ratings, we also analyze results from different perspectives, including in-game success and text generation quality. Experimental findings show that LLMs' conceptual understanding varies across categories and does not consistently align with their general capabilities, highlighting the need for targeted evaluation beyond surface-level performance.

In summary, our contributions are four-fold: (1) we propose CK-Arena, a benchmark for conceptual understanding in interactive multi-agent settings; (2) we develop scalable datasets used in CK-Arena for concept representation, differentiation, and connection, with a simple and expandable data construction process; (3) we conduct a large number of experiments on several large-scale models, and obtained knowledge preferences and behavior preferences of specific large-scale models through qualitative and quantitative analysis; and (4) we establish a scoring system to integrate fine-grained indicators into a comprehensive score, and launch a leaderboard for tested large models.

# 2 RELATED WORKS

**Benchmarks for Conceptual Knowledge Reasoning.** Commonsense reasoning benchmarks play an important role in assessing the capabilities of Large Language Models (LLMs). Widely used

benchmarks such as Story Cloze Test (Mostafazadeh et al., 2017), Choice of Plausible Alternatives (COPA) (Roemmele et al., 2011), and HellaSwag (Zellers et al., 2019) largely rely on static formats like multiple-choice questions or binary judgments. While effective for evaluating factual recall and superficial understanding, these static formats do not fully reflect real-world interactive scenarios. More recent benchmarks, including MMLU (Hendrycks et al.), CMMLU (Li et al., 2024), BIG-Bench (Srivastava et al., 2022), and HELM (Liang et al.), have introduced tasks such as logical reasoning, cloze tests, and multi-turn Q&A to expand the scope of evaluation. Although these efforts represent progress toward more interactive assessments, they still focus predominantly on factual recall and task-specific reasoning, offering limited insight into how well LLMs understand and manipulate conceptual knowledge boundaries in evolving contexts. In contrast, CK-Arena is designed to explicitly evaluate conceptual mastery by immersing LLMs in interactive, multi-agent gameplay that requires real-time understanding of semantic boundaries.

**Game-based Evaluation.**  Multi-agent Games provide a unique platform for evaluating AI capabilities, offering interactive and dynamic environments that differ from traditional benchmarks built on static datasets. They have been used to measure various skills, including environmental perception and planning in exploratory games (Wang et al., 2023; Wu et al., 2023), strategic decision-making in competitive games (Feng et al., 2023; Ma et al., 2024), team collaboration in cooperative games (Agashe et al., 2023; Mosquera et al., 2024), and social interaction and language comprehension in communication games (Light et al.; Qiao et al., 2023; Wu et al., 2024). Compared to static evaluations, game-based benchmarks offer more realistic interaction scenarios that better mimic real-world decision-making.  However, many game benchmarks rely on fixed formats and rules, resulting in gameplay that is highly similar across multiple testing rounds and limiting their evaluation scope. Undercover (Xu et al., 2024) stands out because its interchangeable word pairs generate varied content within the same structure. Although prior work has used Undercover as a benchmark (Xu & Zhong, 2025; Dong et al., 2024; Wei et al., 2025), these studies primarily explored method development and decision-making, without leveraging its unique potential for evaluating conceptual understanding.  CK-Arena fills this gap by integrating concept-based reasoning within multi-agent interactions, allowing LLMs to explore and articulate conceptual relationships dynamically, mirroring real-world cognitive processing.

## 3  CK-ARENA: CONCEPTUAL KNOWLEDGE ARENA

This section introduces the construction of CK-Arena, detailing the choice of the *Undercover* game as the evaluation paradigm, the metrics employed to capture different dimensions of model performance, and the overall workflow for building, running, and analyzing the evaluation. Together, these components establish CK-Arena as a rigorous and scalable framework for uncovering both the strengths and limitations of LLMs in conceptual knowledge.

### 3.1  THE UNDERCOVER GAME FOR EVALUATION

**Game Rule.** CK-Arena is built on the multi-agent language game *Undercover* (Xu et al., 2024), which is originally designed to test the players' reasoning and strategic communication abilities. In the game, players are assigned either as "civilians" who are the majority of the players and know a common word, or as "undercover" who are given a different but related word. Note that each player is informed of their assigned concept word but remains unaware of their team identity or the concepts held by others. Through rounds of description, players must identify who the undercover agents are while undercover agents try to remain undetected by providing descriptions vague enough to seem plausible

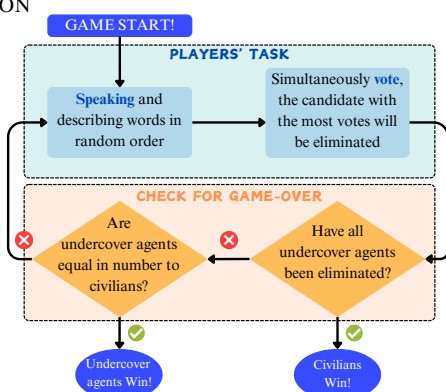

Figure 2: **Flowchart of gameplay and win conditions in Undercover**. Players alternate between describing their assigned concepts and voting to eliminate suspects. The game ends when undercover agents equal the number of civilians (undercover win) or when all undercover agents are eliminated (civilian win).

without revealing their ignorance of the
civilians' word. After each round, play-
ers participate in a voting process to eliminate the individual they suspect to be an undercover agent. The game concludes under one of two conditions: (1) if all undercover agents are eliminated, the civilians win; (2) if the number of civilians and undercover agents is equal, the undercover agents win. The flowchart of gameplay is illustrated in Figure 2.

**Data Statistics**  The dataset we provided contains a total of 529 English pairs of concepts, including 220 concrete noun pairs, 100 abstract noun pairs, 109 adverb pairs, and 100 verb pairs. After initial experimental attempts, we concluded that concrete noun pairs are more suitable for our experimental setup and overall research questions. Therefore, for the specific experiments, we selected 12 different categories from the 220 concrete noun pairs. These categories consist of concrete noun pairs that are closest to our daily life and conversational contexts.

**Why Use Undercover to Evaluate?**  To illustrate the effectiveness of the *Undercover* game in CK-Arena, consider an example where the concepts *football* and *basketball* are assigned to players, with *basketball* designated as the undercover concept. During the speaking phase, the undercover player must analyze the descriptions provided by others about *football*, identify shared attributes, and strategically describe *basketball* in a way that overlaps with common features, such as *"This is a ball-shaped sports equipment"* or *"This sport is played by two teams."* This task requires more than superficial word associations or token co-occurrence. It calls for understanding the similarities and differences between concepts. A model that fails to capture these relationships and relies on shallow generation risks exposing its undercover role and being eliminated. With its emphasis on conceptual understanding, interactive dynamics, and scalable coverage, CK-Arena provides a rigorous benchmark for evaluating LLMs' understanding of conceptual knowledge.

## 3.2 LARGE LANGUAGE MODELS AS PLAYERS

**Pipeline.**  LLM participates in CK-Arena's evaluation by playing multiple rounds of games as a player. In our configuration, we set up 6 players for each game, consisting of 4 civilians and 2 under-cover agents. The game begins with an initialization phase in which players are randomly assigned roles: civilians receive a primary concept, while undercover agents are given a similar but distinct concept. During gameplay, players take turns producing statements that describe their assigned concept while also attempting to identify potential undercover agents or civilians. Specifically, a player's task in the game involves two main components: (1) leveraging partial feature descriptions provided by other players, together with the prior knowledge that the unknown concept is semantically related to the known concept, in order to make inferences about the unknown concept; and (2) retrieving and associating relevant features of their assigned concept based on the given strategic guidelines, and then constructing statements that are related to features of the concept for their turn of speech. These two steps engage the model in processing both concept-to-concept relations and concept-to-feature mappings, thereby providing a strong reflection of its degree of conceptual understanding.

**Prompt Design.**  To ensure effective communication and role-specific behavior, we construct tailored prompts for LLM-based agents in CK-Arena. The prompts include a comprehensive system prompt that provides game rules, input-output format guidelines, specific task instructions, basic strategic guidance, and example descriptions. In addition, each player receives a contextualized user prompt containing information about their assigned concept, historical statements, and analytical insights from previous rounds. Since CK-Arena is designed to evaluate conceptual mastery, we restrict players' strategic space with clear action guidelines to avoid confounding effects from uncontrolled reasoning and decision-making.

## 3.3 EVALUATE THE PERFORMANCE OF PLAYERS

**Data Preparation.**  The selection of concept pairs is crucial to the effectiveness of the *Undercover* game in CK-Arena. We constructed a dataset of semantically related concept pairs spanning a wide range of categories. The dataset underwent pilot screening to ensure two main properties: (1) Semantic proximity: concepts are sufficiently similar to create challenging gameplay yet distinct

Table 1: **Evaluation metrics for CK-Arena.** Detailed breakdown of the metrics used to assess LLM performance in interactive gameplay.

| Metric | Formula | Symbol Definitions |
|---|---|---|
| **Win Rate (WR)** | $\text{WR} = \frac{G_w}{G_t}$ | $G_w$: Number of games won by the player
$G_t$: Total number of games played by the player |
| **Survival Rate (SR)** | $\text{SR} = \frac{R_s}{R_t}$ | $R_s$: Number of rounds the player survived
$R_t$: Total number of rounds in all games |
| **Novelty** | $\text{Nov}(s_i) \in [0, 1]$ | $s_i$: Current statement
$\text{Nov}(s_i)$: Degree of new information in statement $s_i$ compared to previous statements |
| **Reasonableness** | $\text{Rea}(s_i, c) \in [0, 1]$ | $s_i$: Current statement          $c$: Target concept
$\text{Rea}(s_i, c)$: Logical coherence between statement $s_i$ and concept $c$'s properties |

enough for meaningful differentiation; (2) Descriptive clarity: concepts are expressive enough to enable smooth interactions during the game.

The final dataset contains 529 English concept pairs spanning different parts of speech and semantic categories. Detailed statistics are provided in Appendix D, and the source files are available in our project repository. Furthermore, users can freely construct datasets in professional knowledge domains they wish to evaluate, which demonstrates CK-Arena's scalability. Appendix E provides concrete extension examples and guidelines.

**Evaluation Metrics.** To comprehensively evaluate model performance in CK-Arena, we distinguish between two categories of metrics:

(A) *Player-level metrics* capture overall outcomes across games through two measures: Win Rate (WR), which reflects the proportion of games won and indicates effectiveness in fulfilling assigned roles; and Survival Rate (SR), which measures rounds survived before elimination, evaluating players' ability to navigate social dynamics and avoid suspicion.

(B) *Statement-level metrics* assess individual response quality during gameplay, reflecting conceptual mastery at finer granularity. Both metrics use a 0-1.0 scale: Novelty measures new information introduced compared to previous descriptions, promoting creative exploration while preventing repetition; and Reasonableness assesses logical coherence between statements and concept properties, ensuring meaningful discourse. Statements below either threshold trigger automatic elimination.

**Large Language Models as Judges.** To meet the extensive knowledge demands of diverse topics, we adopt a multi-judge pipeline: strong LLMs from different families first produce independent assessments using prompts aligned with the evaluation framework in Section 3.3, where each dimension is defined together with scoring rubrics and worked examples to ensure round-by-round consistency. In order to prevent instability caused by LLMs as judgments, we have set up a manual team to review and adjust some scores based on LLMs' analysis process and relevant open source knowledge bases Miller (1995); wik (a;b). Specifically, 3.1% of the scores were manually calibrated. Once a sufficient volume of annotated data has been collected, the judging process can be further automated, as described below.

**Efficient Metric Assessment.** Considering the high cost and inefficiency of manual expert review, as well as the substantial time and financial burden of relying on powerful LLMs for evaluation, we sought to automate the scoring of two statement-level metrics that directly affect game progress by eliminating players. For *Novelty*, successive descriptive statements within each game can be vectorized, and their similarity (e.g., via cosine distance) can be computed. This provides a direct

Table 2: **Classification performance of *Qwen-3-8B-ckR* as judge.** Our *Qwen-3-8B-ckR* judge on the evaluation set: near-perfect accuracy and $F_1$ score, demonstrating that the fine-tuned model reliably replicates human expert assessments for maintaining game operation.

| Class | Precision | Recall | F1-score | Support |
|---|---|---|---|---|
| 0 | 1.00 | 0.93 | 0.96 | 29 |
| 1 | 0.99 | 1.00 | 1.00 | 271 |
| Accuracy | | 0.99 | | 300 |

quantitative approximation of novelty, where a low similarity score indicates novel contributions.

For *Reasonableness*, the game's elimination mechanism can be viewed as a binary classification task. We tested traditional machine learning classifiers, smaller language models (e.g., MiniLM), and other methods, but these approaches failed to capture conceptual and feature-level relationships. In contrast, *Qwen-3-8B* leveraged the associative reasoning ability of LLMs and achieved 92% accuracy in reasonableness judgment, though its outputs still diverged from human-designed evaluation criteria. To improve alignment, we fine-tuned *Qwen-3-8B* on about 2,000 structured descriptive samples from our experiments, producing *Qwen-3-8B-ckR*. In Table 2, *Qwen-3-8B-ckR* reaches 99.3% accuracy on the test set, and we adopt it as the reasonableness judge.

**Result Collection and Analysis.** CK-Arena integrates comprehensive data collection throughout each gameplay session. Every game instance generates a structured JSON record containing metadata (ID, timestamp, selected concepts), player details (IDs, LLM models, roles, assigned concepts), and judge specifications. The system logs the complete history of player statements with evaluation scores for novelty and reasonableness, along with vote records, elimination outcomes, and game-level statistics that capture overall performance and decision-making patterns. Users can choose to retain only statements and votes or the full reasoning process. All data are organized by rounds, enabling multi-dimensional analysis of interactions and decisions. Automated scripts further aggregate results across instances, producing statistical summaries and visualizations of indicators such as decision quality, elimination accuracy, and statement metrics.

**Unified Rating System.** To move beyond single-batch evaluations, we introduce a robust rating system that supports repeated assessments across multiple batches using quantitative indicators to systematically track model performance in CK-Arena. Because player behavior spans multiple dimensions, including win rates, survival rates, voting accuracy, and other indicators, the system provides a unified framework to capture overall capability. Specifically, we implement a team-based Elo rating system tailored to CK-Arena, where each player's rating is dynamically updated based on game outcomes, performance metrics, opponent and teammate strength, and experience-dependent volatility factors (Elo & Sloan, 1978).

For each player $i$ in game $g$, we compute a composite performance score $S_i^g$ as a weighted combination of multiple performance indicators:

$$S_i^g = \alpha \cdot W_i^g + \beta \cdot SR_i^g + \gamma \cdot VR_i^g$$

where $W_i^g \in \{0,1\}$ represents the binary win/loss outcome, $V_i^g \in [0,1]$ denotes the survival rate, and $VR_i^g \in [0,1]$ represents correctly voting rate. In our experiments, we set $(\alpha, \beta, \gamma) = (0.75, 0.15, 0.10)$.

To account for differing uncertainty in rating estimates between novice and experienced players, we group games into batches of 12 and apply an experience-dependent K-factor that decays by batch rather than by individual game count. This batching reflects the game design: different topic words may introduce systematic variation, and batch-based evaluation balances rating adjustments across diverse themes. The K-factor is defined as

$$K(n) = K_{min} + (K_{max} - K_{min}) \cdot \exp\left(-\frac{\lfloor n/12 \rfloor}{\tau}\right)$$

where $n$ represents the number of games played and we set $K_{max} = 60$, $K_{min} = 5$, $\tau = 2.5$. This formulation ensures high volatility for new players ($K \approx 60$ at $n = 0$) while stabilizing ratings for experienced players ($K \approx 5$ at $n \geq 140$).

We conducted an analysis of the results in preliminary experiments and observed an inherent role bias in *Undercover*: under the 2 versus 4 setting with our defined prompts, civilians are consistently more likely to win than undercover agents (with an average win rate of approximately 66.7%). This phenomenon has also been reported in several other studies related to *Undercover* (Dong et al., 2024; Xu & Zhong, 2025). To correct for this role-induced imbalance, we introduced an adjustment in the computation of expected performance. Specifically, during the calculation of expected performance, we add a temporary Elo offset of +120 to the stronger side (*i.e.*, the civilian role). This adjustment ensures that players of equal skill level have comparable rating update opportunities regardless of whether they play as civilians or undercover agents. The detailed derivation and justification of the 120-point offset are provided in the Appendix D.

Table 3: **Performance comparison in CK-Arena.** Results are reported separately for the *Civilian* and *Undercover* roles. WR denotes *Win Rate*, and SR denotes *Survival Rate*. Both serve as indicators of in-game performance, where higher values reflect stronger capability in fulfilling role objectives. Reasonableness measures the logical consistency of statements with the target concept, while Novelty evaluates the degree of new information introduced. We show how models balance these factors, with Qwen2.5-72B leading in reasonableness, GPT-4o showing strong civilian win rates, and Gemini-2.0-pro-exp excelling in novelty. The best values are in **bold** and the second-best are underlined.

| LLM | Role | Performance Metrics | | | |
|---|---|---|---|---|---|
| | | WR ↑ | SR ↑ | Reasonableness ↑ | Novelty ↑ |
| Qwen2.5-72B | Civilian | 0.6847 | **0.7207** | 0.9593 | 0.6676 |
| | Undercover | **0.3636** | **0.2955** | **0.9737** | 0.7051 |
| GPT-4o | Civilian | **0.6854** | 0.6629 | 0.9678 | 0.6693 |
| | Undercover | 0.3485 | 0.2273 | 0.9614 | 0.7429 |
| DeepSeek-V3 | Civilian | 0.6814 | 0.6637 | 0.9470 | 0.8248 |
| | Undercover | 0.3571 | 0.2857 | 0.9537 | 0.8220 |
| LLaMA-3.3-70B-instruct | Civilian | 0.6702 | 0.6596 | 0.9663 | 0.8072 |
| | Undercover | 0.3279 | 0.1803 | 0.9678 | 0.8083 |
| Gemini-2.0-pro-exp | Civilian | 0.6636 | 0.6545 | 0.9667 | 0.8259 |
| | Undercover | 0.3111 | 0.2889 | 0.9652 | **0.8391** |
| Claude-3-5-Haiku | Civilian | 0.6408 | 0.6214 | 0.9494 | 0.7633 |
| | Undercover | 0.2692 | 0.1923 | 0.9273 | 0.8061 |

## 4 EXPERIMENTS

In this section, we describe the experimental setup and present the main findings, including the evaluation of large models in CK-Arena. Our experiments follow two steps. First, we perform baseline evaluations on six widely-used LLMs from different families in controlled 6-player games, focusing on statement-level performance, conceptual understanding, and role-specific metrics. Second, we construct a scalable leaderboard, where additional LLMs are benchmarked against some of the six baseline models, which serve as anchors in our unified rating system. This allows us to quantify relative strength across a broader set of models. We report both quantitative and qualitative analyses to ensure the reliability of the results.

The testing data consists of 464 game instances across twelve concept categories: *food, landforms, animals, artifacts, tools, people/social, plants, sports, stationery, electronics, clothing*, and *sundries*. During gameplay, a total of 6112 conceptual feature descriptions are generated. Additional results are provided in Appendix D and Appendix E.

### 4.1 RESULTS ON THE 6-PLAYER GAME

**Experimental Setting.** We evaluate six widely used LLMs from different families, including *Claude-3-5-Haiku* (Anthropic, 2024), *GPT-4o* (Hurst et al., 2024), *Gemini-2.0-Pro-Exp* (Team et al., 2023), *DeepSeek-V3* (Liu et al., 2024), *LLaMA-3.3-70B* (Grattafiori et al., 2024), and *Qwen2.5-72B* (Bai et al., 2023). In addition, *GPT-4.1-2025-04-14* (OpenAI, 2025a) and *Claude-3-7-Sonnet-20250219* (Anthropic, 2025a) are selected as the LLM-based judges to score all statements across statement-level metrics as references. Following data collection, a human expert panel then reviewed all statements, taking into account both the LLMs' scores and relevant reference knowledge, and determined the final scores.

**Performance Comparison.** Table 3 summarizes baseline model performance in CK-Arena. Civilian win rates are consistently higher than undercover win rates, showing that the undercover role is more challenging because it requires concealing one's assigned concept while simultaneously inferring shared features with the civilian concept. All LLMs performed well at reasonableness. This is partly attributed to the threshold elimination mechanism applied during the evaluation, which filters

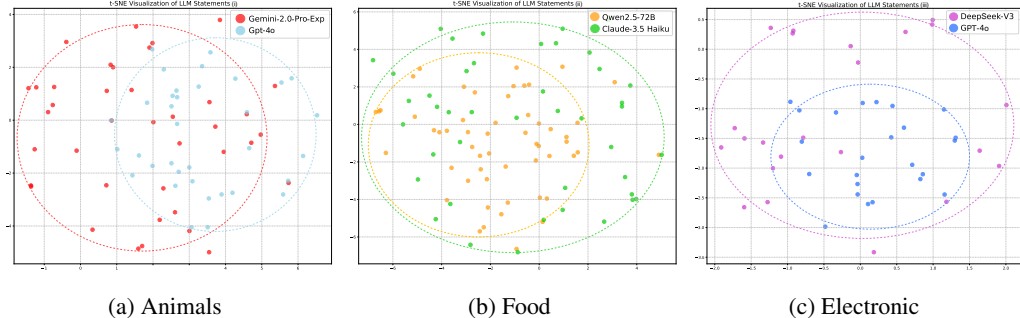

(a) Animals       (b) Food       (c) Electronic

Figure 3: **t-SNE visualizations of LLM statements across concept categories.** Each plot shows model outputs for (a) animals, (b) food, and (c) electronics. Repetitive descriptions, reflecting shallow understanding, appear as tightly clustered points, whereas richer knowledge produces more dispersed distributions. The visualizations also indicate that different LLMs center their descriptions on different focal aspects of a concept, suggesting variation in how conceptual knowledge is represented.

out low-scoring statements before final analysis. High reasonableness scores also indicate that current LLMs are capable of understanding tasks and generating structured language descriptions based on basic knowledge. Novelty is more nuanced: strong models such as *Qwen2.5* and *GPT-4o* often score lower, as excessive novelty risks revealing the undercover identity, while repetition can also lead to elimination. Effective play depends on balancing precision and originality, highlighting the ability to express subtle conceptual differences without being overly novel or trivially repetitive. CK-Arena is designed to capture this balance, making it a meaningful test of conceptual reasoning.

**Semantic Dispersion as a Proxy for Conceptual Depth.** We embed LLM-generated statements and compare them using dimensionality reduction and visualization. Given the same number of descriptions for a concept, shallow understanding typically leads to repetitive phrasing, which appears as tightly clustered points in the t-SNE plot, whereas deeper knowledge produces more dispersed patterns. The visualizations reveal systematic differences in how models generate conceptual descriptions under the same topic. Figure 3(a) shows that *Gemini-2.0-pro-exp* and *GPT-4o* emphasize different aspects of the same concept, reflecting variation in conceptual focus. Figures 3(b) and (c) further demonstrate differences in clustering degree, with some models producing narrow clusters

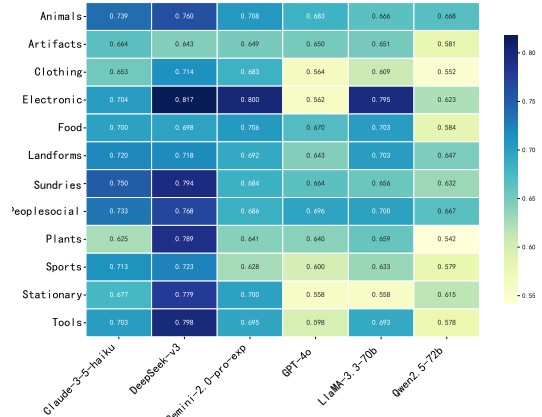

Figure 4: **Relevance scores of different LLMs across various categories**. The heatmap shows how well model statements align with target concepts, where darker colors indicate higher relevance.

and others spreading more broadly across the semantic space. This indicates the variation in focus and the degree of dispersion in LLM-generated conceptual associations.

**Statement–Concept Relevance Heatmap.** In addition to the metrics introduced in Section 3.3, we also evaluate *Relevance*, where LLM judges score each player's statements based on alignment with the target concept. High scores correspond to specific, tightly linked descriptions that help civilians detect the undercover agent, while low scores indicate vague or overly broad statements that could fit multiple concepts. This metric reflects the strategic tension of the game: civilians gain from precise descriptions, whereas undercover agents may opt for broader ones to avoid detection.

Figure 4 presents the relevance scores of different LLMs across conceptual categories. Both the highest-scoring *DeepSeek-V3* and the lowest-scoring *Qwen2.5-72B* also achieve strong win rates, showing that higher relevance does not necessarily lead to better game performance. At the same

time, scores remain relatively consistent across categories. This suggests the chosen concepts are similarly describable, which helps ensure that the benchmark evaluates models fairly rather than being driven by category-specific difficulty.

## 4.2 THE SCALABLE LEADERBOARD IN CK-ARENA

**Experimental Setting.** We construct a leaderboard of LLMs as players in CK-Arena. *DeepSeek-v3* and *Qwen2.5-72B* serve as anchor models, providing stable baselines for comparison. We then benchmark additional LLMs, including *GPT-5* (OpenAI, 2025b), *GPT-oss-120b* (OpenAI et al., 2025), *DeepSeek-v3.1* (DeepSeek, 2025), *Claude-opus-4.1* (Anthropic, 2025b), *kimi-k2-instruct* (Team et al., 2025), *Qwen-plus* (Yang et al., 2025), *Ernie-4.5-300b-a47b* (Baidu-ERNIE-Team, 2025), and *Gemini-2.5-flash-preview* (Google, 2025). Each model receives identical prompts and plays at least 60 rounds against the anchor models to ensure rating stability and reliability.

**Results from Unified Rating System.** Each new model plays more than 60 rounds against the anchors, allowing Elo score fluctuations to stabilize under the experience-dependent $K$-factor schedule. To control for ordering effects, we also reverse the sequence in which models are introduced. Forward and reverse evaluations produce the same ranking across all 14 LLMs, with the maximum Elo difference for any model being only 1.72 and a Pearson correlation of 0.99 between leaderboards. This high consistency shows that evaluation order does not affect fairness and validates the design choice of using experience-dependent $K$-factors and anchor baselines. We also added human baseline to benchmark to reflect the general human level of conceptual knowledge application to a certain extent. However, we need to emphasize that humans will have some disadvantages compared with LLM in the large-scale evaluation of undercover, because the multiple undercover games involved in CK-arena require participants to have a profound understanding of all aspects of knowledge, but it is difficult for normal humans to distinguish these details. For details, please refer to Appendix E.

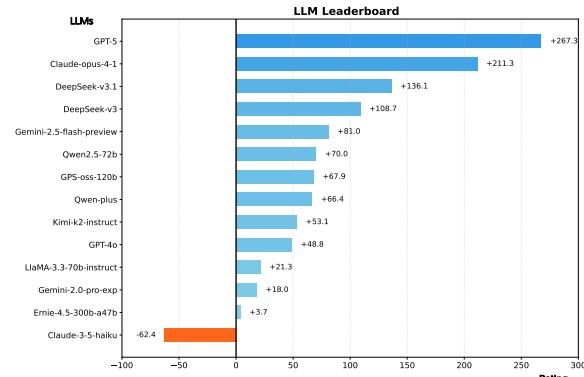

Figure 5: **Leaderboard of LLMs in CK-Arena.** Each player starts with an initial rating of 0. After stabilization, a player consistently defeating 0-rated opponents converges around 420, which serves as a reference for strong performance. The leaderboard highlights relative differences across 14 evaluated LLMs.

Using this framework, we compile a comprehensive leaderboard of all evaluated LLMs (Figure 5). Within model families such as *DeepSeek* and *Qwen*, performance gaps between newer and older versions are relatively small, and both remain behind top-tier models like *GPT-5*. These results suggest that iterative improvements within some families are not sufficient to close the performance gap, an aspect that CK-Arena makes visible.

## 5 CONCLUSION

We present CK-Arena as a benchmark for evaluating the conceptual knowledge and understanding of LLMs through interactive, multi-agent gameplay. Built on the *Undercover* game, it provides a scalable and dynamic environment where models engage with associations, similarities, and differences between concepts, an ability that traditional static benchmarks often overlook. Our experiments show that conceptual understanding varies across categories and does not consistently align with general benchmark performance, indicating that skills such as coding or mathematics do not necessarily translate into stronger conceptual understanding. CK-Arena addresses this gap with a systematic and extensible framework for assessing conceptual knowledge, and it serves as a starting point for future benchmarks that seek to capture more human-like, semantically grounded understanding in LLMs.

**Limitations and future directions.** We acknowledge that CK-Arena has several limitations. As an early effort in evaluating conceptual understanding, it still depends on strong LLMs and some manual review for judging, enforces strict response formats that may penalize formatting errors, and is restricted to English, which limits cross-linguistic evaluation. These limitations point to clear directions for future work, such as developing more robust automated judging mechanisms, enhancing response handling, and extending the benchmark to multiple languages. Addressing these challenges will improve the scalability, reliability, and inclusiveness of CK-Arena, strengthening its role as a foundation for conceptual understanding evaluation.

## ETHICAL STATEMENT

This research was conducted following established ethical guidelines for AI research. Our benchmark CK-Arena evaluates AI systems' conceptual knowledge without collecting or processing any personally identifiable information. All concept pairs used in our experiments were carefully curated to ensure they do not contain harmful, offensive, or culturally insensitive content. The experiments involving multiple large language models were designed to analyze their capabilities in understanding conceptual boundaries without any deception or manipulation techniques.

## REPRODICIBILITY STATEMENT

We provide all resources necessary to reproduce our work. The complete code, dataset, and training data used in our experiments are released together with this paper. The prompts used, parameter settings for LLM utilization, and hyperparameter configurations for fine-tuning the large model have all been disclosed in Appendix D. In addition, we include a scalability demonstration and an example in Appendix E to facilitate replication.

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

## A  FUTURE WORKS

In the future, we plan to extend CK-Arena in several key directions: (1) Expanding the Concept Pair Dataset: We aim to increase the diversity of concept pairs by introducing more categories and refining the quality of selections, thereby building a more comprehensive knowledge network for evaluation. (2) Multilingual Extension: Adapting CK-Arena to support multiple languages holds significant potential. Different languages are deeply tied to unique cultural knowledge and conceptual representations, which can reveal cross-linguistic differences in conceptual understanding. (3) Diversifying Agent Forms: Beyond standard LLM-based agents, we intend to incorporate specialized language models trained in specific knowledge domains to serve as judges, and even explore scenarios where LLM-based agents interact and compete alongside human participants. Furthermore, the rich set of statements generated during CK-Arena gameplay represents a valuable resource. These concept-driven descriptions can form a semantic norm, potentially serving as raw data for training concept-aware models, such as Large Concept Models (LCMs). Although the current dataset is functional, we aim to further enhance the automation process and evaluation system to transform this data into a high-quality, structured dataset. This would enable more effective training and evaluation of models designed for conceptual understanding and knowledge-based tasks.

## B  LIMITATIONS

Despite its contributions, the CK-Arena benchmark also has several limitations that are worth considering. First, during the initialization stage, the LLM serving as the judge must be a powerful and knowledgeable model, and the final scoring still requires manual team review. While our automated pipeline shields end users from these concerns, researchers may incur additional time and financial costs if they prefer to define their own judging criteria rather than adopting CK-Arena's defaults. Second, our framework places strict requirements on the format of LLM responses (e.g., JSON). Although we implement parsing and error-handling mechanisms, models may still be penalized for formatting issues rather than genuine gameplay mistakes. This may require users to regularly check game logs, identify abnormal responses, and supplement more response handling mechanisms. Third, all evaluations are conducted exclusively in English, which may introduce language-specific biases and constrain cross-linguistic insights into conceptual mastery. Addressing these challenges will be crucial for improving the scalability and inclusiveness of the benchmark.

## C  USE OF LLMS

In the course of this work, we employed Large Language Models (LLMs) in two ways. First, LLMs (specifically *Claude Sonnet 4*) were used during manuscript preparation for grammar checking, text polishing, and improving the clarity of academic writing. Second, in the early stages of literature review, we utilized the "deep research" function of LLMs to obtain a broader and more comprehensive overview of related works. These applications were limited to auxiliary support and did not influence the design, implementation, or analysis of CK-Arena.

## D  IMPLEMENTATION DETAILS

**Detailed Data Statistics**   The dataset we provided contains a total of 529 English pairs of concepts, including 220 concrete noun pairs, 100 abstract noun pairs, 109 adverb pairs, and 100 verb pairs. After initial experimental attempts, we concluded that concrete noun pairs are more suitable for our experimental setup and overall research questions. Therefore, for the specific experiments, we selected 12 different categories from the 220 concrete noun pairs. These categories consist of concrete noun pairs that are closest to our daily life and conversational contexts. All of those concepts can be considered with rich and clearly describable features. We believe that starting with these concept pairs can more reliably and steadily complete our experiments and yield preliminary results. In the future, we will further explore the other words.

**Experimental Settings**   For the evaluation of LLMs as players, all models were used in a zero-shot setting without task-specific fine-tuning. We only specified the input prompts, without any additional

hyperparameters, such as temperature, top-p, and so on. To ensure fairness and reproducibility, we employed the default API settings for each model, consistently choosing the most recent stable release available at the time of the experiments.

For the training of qwen-3-8b-ckR, we fine-tuned the model using LoRA adaptation with the following hyperparameter configuration: 5 training epochs, a learning rate of 1e-4, batch size of 16, linear learning rate scheduler, validation every 50 steps, maximum sequence length of 8192, warmup ratio of 0.05, and weight decay of 0.01. LoRA-specific settings included rank 8, alpha 16, dropout 0.1, and applying adaptation to all target modules. The dataset used for training, along with preprocessing details, is fully released in the accompanying code repository to ensure reproducibility.

**Connection of CK-Arena with Existing Benchmarks.** We think that CK-Arena offers distinct yet complementary value in LLM evaluation tasks.

Fundamental Differences in Evaluation Focus: Traditional benchmarks like MMLU primarily assess factual recall and static knowledge retrieval through multiple-choice questions. In contrast, CK-Arena evaluates dynamic conceptual understanding in interactive contexts. For example, while MMLU might ask "Which of the following animals is a primate?", CK-Arena requires models to articulate the distinguishing features between closely related concepts (e.g., monkeys vs. apes) and navigate the semantic boundaries dynamically based on partial information from other agents.

Why Static vs. Dynamic Evaluation Matters: Our preliminary analysis suggests that strong performance on traditional benchmarks doesn't necessarily translate to effective conceptual boundary navigation. For instance, a model might correctly identify that both soccer and basketball are sports (factual knowledge) but struggle to strategically describe one while concealing its identity when the other is the majority concept (conceptual understanding + strategic reasoning). This highlights that knowing facts about concepts differs from understanding their relational structures and boundaries.

We also point out that CK-Arena does not aim to replace existing benchmarks but to fill a critical gap in evaluating interactive conceptual understanding. Traditional benchmarks excel at measuring breadth of knowledge, while CK-Arena probes depth of conceptual understanding in realistic social contexts. The differences in results reflect that multi-agent interaction requires different cognitive processes than isolated question-answering.

**Scalability Demonstration.** In order to explain the scalability of CK-Arena, we provide a specific example in this section to help researchers who need to build their own datasets test LLM's knowledge mastery in specific fields. We will divide this task into three steps:

Firstly, researchers need to construct concept pairs related to evaluation knowledge within their field (for example, by describing the similarities and differences between alcohol lamps and flame spray guns to explore the knowledge of middle school chemistry experiments). Users may also need to adjust prompts if they wish to have their own rating criteria. Then, users need to conduct at least 60 pre-experiments using models with comparable performance (or one model as all players) and game settings of their own choice (such as number of players, rounds, etc.) to obtain role bias calibration values. Specifically, the concepts in the newly constructed dataset may have inconsistent similarities, which can lead to role bias in the game. For example, if two concepts are very similar, it is obvious that undercover characters are easily mixed up with civilian characters; On the contrary, the undercover character finds it difficult to move forward. Therefore, it is necessary to determine role bias through pre-experiments and use temporary scores to balance this bias. The third step is for users to repeat the game multiple times until the K-value stabilizes, in order to obtain a performance analysis among the LLM players participating in the game.

Then, here comes the example. Due to the fact that most concepts that contain broadly descriptive features are nouns, our specially designed prompt template is not suitable for evaluating verbs or other parts of speech. Therefore, we carried out a complete extension proces. First, we built a verb word pair dataset, and then adjusted part of the content in the prompt to help players better participate in the game, and judges more standardized. The following are the added parts:

*- Nature of the action: Such as the type characteristics of the action. - Relationship of action: Such as the characteristics of the subject and object involved. - Usage scenarios: Such as the environmental characteristics and cultural background where the action occurs. - Concluding effects:*

*The consequences and impacts brought about by the action. - Emotional impact: The emotional overtones, moral implications, and social attributes involved in the action.*

The experimental results regarding verbs can be viewed in section E. During our testing, the API call cost for reviewing a single game was approximately \$0.8, while completing a full theme review required around \$40–50. By replacing expensive LLM-based judges with a fine-tuned model, as mentioned in the paper, these costs could be more substantially reduced. In terms of time efficiency, the open-source code provided in this work already supports batch execution of multiple games. Although API calls impose certain speed constraints, our experiments show that running 5 games in parallel does not trigger rate-limit restrictions, allowing most reviews to be completed in only one day.

**Derivation of the 120-Point Elo Offset.** In this paragraph, we derive the 120-point Elo offset used to balance the expected performance between the civilian and undercover roles in the game. The goal is to ensure that players of equal skill levels have comparable rating update opportunities, regardless of their assigned roles.

In the Elo rating system, the expected score $E_A$ of player $A$ against player $B$ is given by:

$$E_A = \frac{1}{1 + 10^{(R_B - R_A)/400}},$$

where $R_A$ and $R_B$ are the ratings of players $A$ and $B$. Although this is a 1v1 formula, in our design, the Elo update first computes the expected outcome based on the win–loss relationship between two teams, and then incorporates each player's individual performance for the actual score adjustment. Therefore, we can treat the two teams as player A and player B, and use the standard formula for derivation.

Empirically, the civilian role has a natural advantage, leading to a baseline win probability of $2/3$ for the civilians against undercover agents of equal skill. To determine the Elo offset that corresponds to this advantage, we solve for the Elo rating difference $x$ that yields an expected score of $2/3$:

$$\frac{1}{1 + 10^{-x/400}} = \frac{2}{3} \tag{1}$$

$$10^{-x/400} = \frac{1}{2} \tag{2}$$

$$-\frac{x}{400} = \log_{10}\left(\tfrac{1}{2}\right) \tag{3}$$

$$x = 400 \cdot \log_{10}(2) \approx 400 \cdot 0.3010 \approx 120 \tag{4}$$

Thus, an Elo difference of approximately 120 corresponds to the observed $2/3$ win rate. To balance the game, we introduce a temporary offset of $+120$ Elo points to the civilian side when computing expected outcomes. This adjustment ensures that, from the model's perspective, the expected probability of winning for both sides is effectively $1/2$, thereby eliminating the systematic role-induced imbalance in rating updates.

## E   MORE EXPERIMENTAL RESULTS

**The stability of the scoring process**   To verify the stability of the scoring process in our LLM-based evaluation framework (and thereby support the reliability and repeatability of evaluation results), we conducted three independent evaluations on the animal group. Based on the outcomes of these evaluations, we calculated key statistical indicators—mean, variance, and standard deviation—for each of the statement-level metrics (Novelty and Reasonableness). The specific statistical data are presented in Table 4. This table reflects the stability of the scoring process: LLM-based assessments already demonstrate strong internal consistency, and with additional human review to adjust specific cases, CK-Arena ensures both reproducibility and robustness of the evaluation framework.

Table 4: Statistical indicators of three independent evaluations on the animal group.

| Metric | Mean | Variance | Std Dev |
|---|---|---|---|
| Novelty | 0.8150 | 0.000203 | 0.0142 |
| Reasonableness | 0.9672 | 0.000042 | 0.0065 |

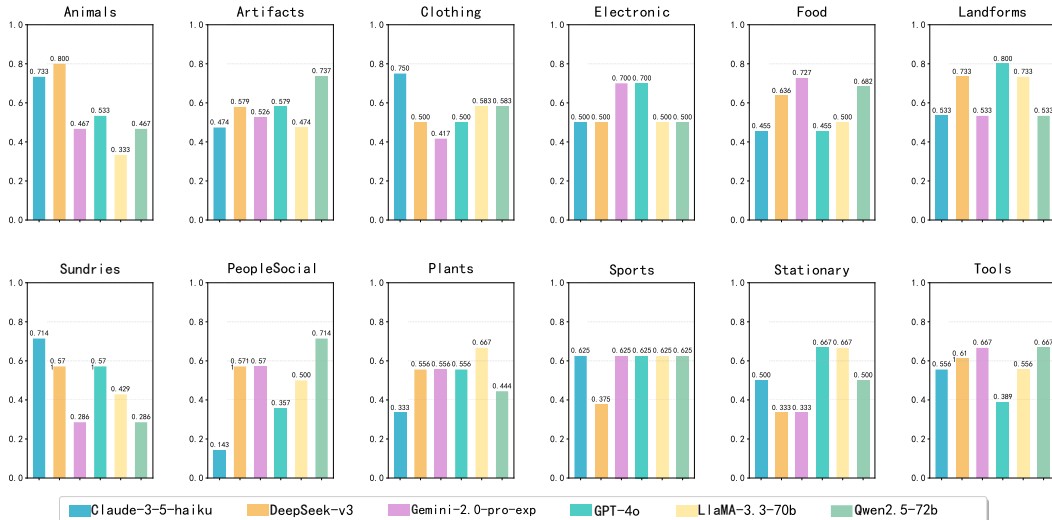

Figure 6: **The win rate performance of six LLMs across** 12 **categories.** A comparative analysis reveals that each model exhibits distinct strengths and weaknesses across different concept categories. These variations are likely influenced by differences in training data, architectural design, and optimization strategies specific to each model. The analysis reveals models' focus areas, knowledge gaps, and insights for improving conceptual understanding.

**Win Rate by Different Categories.** Figure 6 illustrates the win rate performance of various LLMs across different conceptual categories. The results highlight clear strengths and weaknesses for each model. For example, DeepSeek-V3 achieves the highest win rate in the animal category, reaching 80%, indicating strong domain-specific understanding. Similarly, *GPT-4o* excels in the landmark category with a win rate of 80%, reflecting its grasp of geographical concepts. In contrast, *Claude-3-5-Haiku* demonstrates a notably low win rate of just 14.3% in the social category, suggesting limitations in handling social context. These performance differences are likely influenced by the models' training datasets and optimization strategies, highlighting domain-specific expertise and gaps in conceptual understanding.

Table 5: **Win rate (WR) and Survival rate (SR) comparison of baseline large language models in CK-Arena.** Results are reported separately for *Civilian* and *Undercover* roles.

| LLM | Role | Performance Metrics | |
| --- | --- | --- | --- |
| | | WR ↑ | SR ↑ |
| DeepSeek-V3 | Civilian | 0.4286 | 0.7391 |
| | Undercover | 0.3750 | 0.5714 |
| Gemini-2.0-exp | Civilian | 0.4762 | 0.4286 |
| | Undercover | 0.3333 | 0.2222 |
| Claude-3-5-Haiku-20241022 | Civilian | 0.6842 | 0.5263 |
| | Undercover | 0.7273 | 0.6364 |
| Qwen2.5-72B | Civilian | 0.5652 | 0.7391 |
| | Undercover | 0.5714 | 0.5714 |
| LLaMA-3.3-70B-instruct | Civilian | 0.5238 | 0.7143 |
| | Undercover | 0.4444 | 0.4444 |
| GPT-4o-2024-11-20 | Civilian | 0.5000 | 0.5455 |
| | Undercover | 0.3750 | 0.3750 |

**Evaluation Based on Verb Vocabulary**    We repeated the baseline experiment, but changed the dataset used for evaluation to one with verb themes. Specific data can be found in our open-source repository as shown in Figure 5. Interestingly, Claude-3.5, which had always performed at the bottom of the original model, actually achieved the highest win rate in this experiment and showed a significant gap compared to other models. Perhaps we can conduct more fine-grained classification and evaluation to explore the reasons for these phenomena.

**Leaderboard with Reasoner Models and Human Baseline**    An intuitive guess is that when large models complete complex language tasks, such as CK-Arena for multi-agent interactions, consuming more tokens for thinking and reasoning will lead to better game performance. Although we use prompts templates to restrict consistent strategies, will using the same strategy guidance in the model lead to significant differences in game outcomes? We supplemented this with ablation experiments and evaluated CK-Arena using common inference models from various families, including o1, DeepSeek-R1, Qwq-plus, and Gemini-2.5-Pro-Thinking. After participating in the same evaluation process, we added the scores of these models to the leaderboard. The results are reassuring: these inference models have some performance differences compared to their original models of the same period and family, but there is no problem with the reasoner being significantly stronger than the original model: DeepSeek-R1 and Gemini-2.5-pro-thinking are even worse than ordinary models in the same series. This further indicates that our benchmark restricts the strategic behavior of the evaluated through prompts and pipelines, and the evaluation focuses entirely on understanding concepts, capturing features, and then expressing them in language.

Another interesting issue is the human baseline. We have added a startup script for human AI confrontation in the project code and collected some evaluation results; however, there are many reasons that prevent us from completing a complete and AI-consistent evaluation process. The main problem is that human knowledge reserves are completely inferior to LLM. This will result in humans triggering more rationality detection mechanisms and novelty monitoring mechanisms during the evaluation process than LLM, leading to the elimination or the inability to describe detailed features when necessary. To provide reproducible reference values under existing conditions, we adopt a "confidence screening" remedial approach: allowing participants to self-evaluate their familiarity after seeing words, and only retaining the matches they consider "familiar" to enter the final statistics. We emphasize that this baseline is a 'lower limit reference' rather than an 'upper limit benchmark', as the sample size is not as same as LLMs' evaluation, and confidence screening may introduce overestimation bias. In the future, we will try to expand the sample pool and introduce an "open book" mechanism (allowing retrieval tools) to reduce the elimination rate caused by differences in knowledge reserves, and design a "human-machine hybrid" evaluation protocol to prevent human language expressions from being voted out due to their incompatibility with the five LLMs.

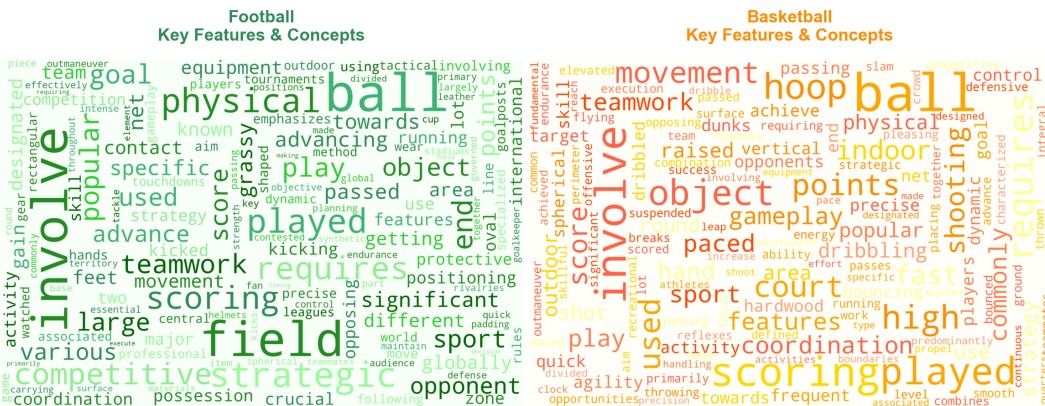

Figure 8: **Word cloud constructed through the game process.** Intuitively reflecting the knowledge content that LLMs are more likely to think of when discussing related topics.

**Reuse Data to Construct Knowledge Graphs**  Since the fundamental goal of the Undercover task is to describe its features around concepts, the game logs recorded during our large models evaluation actually contain a large number of feature descriptions from different perspectives of a concept. After refining and abstracting these descriptions, we can easily construct a knowledge graph that can retrieve feature descriptions with different relevance scores through concept retrieval. Firstly, we have provided integration scripts in the project code that can help users organize game logs into retrievable JSON files to build a complete, unmodified knowledge graph. Edge nodes record the

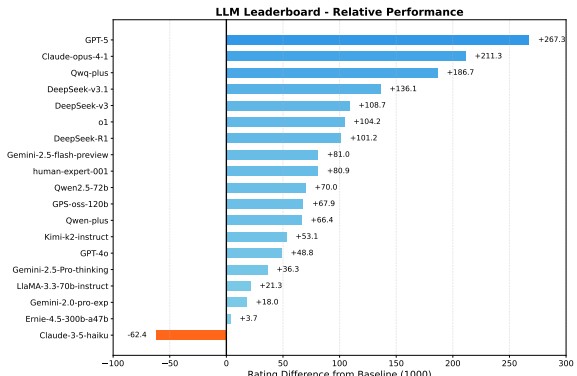

Figure 7: **Leaderboard with Reasoner Models and Human Baseline.**

metric of relevance during the game process, so that fuzzy or specific feature descriptions with different degrees of relevance can be retrieved according to needs. In addition, we can also transform complete feature description sentences into phrases and words. We have provided a simple example here. Figure 9 shows the common and differential features of feature pairs, while Figure 8 reflects what feature content LLM-based agents are most likely to associate with when this concept is mentioned. This knowledge graph cannot guarantee complete accuracy, but it excels in being richer, more generalized, and more associative. Although this article did not systematically validate the downstream benefits of the graph, preliminary observations suggest three potential uses: Provide interpretable "soft definition" completion for the field of data scarcity; As a probe, it can quickly detect the stereotype bias or weakness of a certain concept in a model; Directly injecting into the dialogue system allows open domain replies to have more three-dimensional details at the level of 'what to say'.

**Specific Case Analysis**  We provide various elimination cases to demonstrate the challenges LLM will face in participating in the CK Arena evaluation process, reflecting its required capabilities.

Firstly, as mentioned in the main text, games sometimes require players to describe the vague features of concepts, the common features of two concepts, and sometimes the specific features of individual concepts depending on the specific process and situation. We can see in case 1: In the first round of speeches, everyone was relatively conservative, and the descriptions given by civilian players were also quite broad, which led to confusion during the voting process: the majority of civilians were unable to analyze their identity and opponents from different camps based on existing information, and ultimately one civilian was voted out.

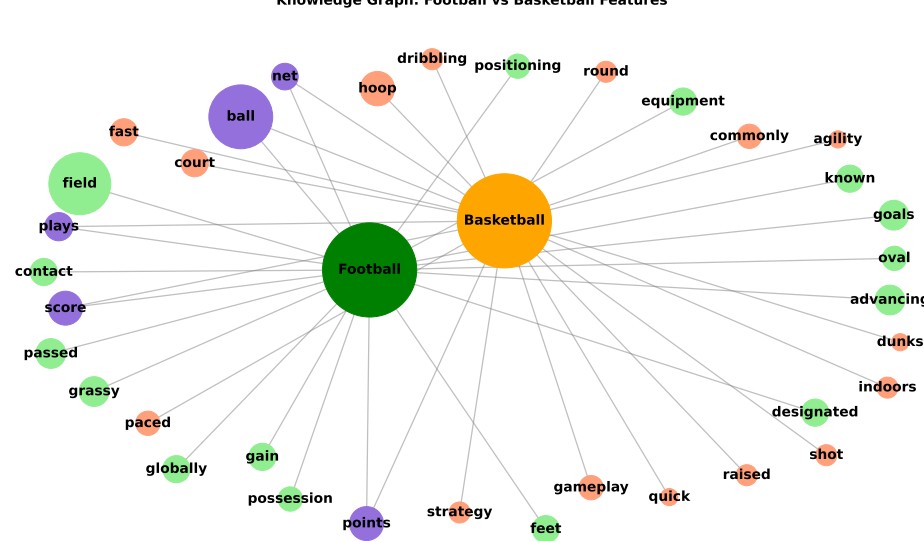

Figure 9: **A simplified version of the knowledge graph extracted through descriptive statements.** The node size reflects the frequency at which features are mentioned. Green is a characteristic of the football concept, orange is basketball, and purple is shared by both.

Looking at case 2, Player 6, as the only undercover agent left in the room, should seek the commonality between crayons and markers to avoid exposing themselves; However, perhaps it was due to the inability to infer the information of Mark Pen and himself as an undercover agent from the statements of the other three players, or perhaps it was because the common features could have been thought of after the first two rounds were finished. Player 6's statements in this round exposed Crayon's unique information, resulting in being voted out.

Two specific cases demonstrate that Undercover does not have a universal problem-solving strategy, as players do not know their identities in advance to determine whether they need to describe themselves more broadly or precisely. This involves analyzing others' broad descriptions, piecing together multiple features to form two conceptual ranges, finding their intersecting and unique parts, and then clarifying their identity and game objectives.

We also provided two cases to explain the importance of Novelty and Reasonability.

In case 3, the undercover agent has actually been exposed, and the citizens have already provided specific descriptions to verify each other's identities. If this round of the game ends normally, the undercover agents should be voted out one by one. However, the descriptions of Player 6 and Player 5 were found to be duplicated, resulting in Player 6 being directly expelled. This also demonstrates the difference between interactive tasks and static testing: if DeepSeek-V3 is asked to name the features of a tiger, it may be able to list a hundred; However, in real tasks, after being told the importance of not repeating others' descriptions, it cannot choose other content to express.

Case 4 showcases the "little smart" attempts of human players: due to the low speaking order of human players in this game, based on the descriptions of whales by the citizens earlier, players have already guessed that the shark they received corresponds to an undercover agent. So the player wants to directly describe the specific characteristics of the whale to blend in with the citizen team. However, this does not align with the starting point of our CK Arena: we hope that high scores reflect a deep understanding of two concepts, rather than deception and strategy. In this case, the automatic judgment process detected that the player's description did not match the word 'shark', so it directly eliminated them.

From this, it can be seen that Undercover is not a simple "guessing word" game, but a collective reasoning experiment that constantly evolves in information gaps, cognitive biases, and real-time

interactions. Only when models or humans can accurately extract features, dynamically read context, control risks, and generate information increments, can they survive in repeated voting.

```
ROUND 1 - STATEMENTS

Player 1 (claude-opus-4-1-20250805) (CIVILIAN - surfboard):
"A piece of equipment that helps people glide across a surface."

Player 6 (gpt-5) (UNDERCOVER - paddleboard):
"Often paired with a leash and carried by hand to the shoreline."

Player 4 (gemini-2.5-flash-preview-05-20) (UNDERCOVER - paddleboard):
"It provides a platform for individuals to balance upon while on water."

Player 5 (ernie-4.5-300b-a47b) (CIVILIAN - surfboard):
"It's commonly used in coastal areas for recreational activities."

Player 3 (qwen2.5-72b) (CIVILIAN - surfboard):
"A flat, elongated board used for movement and balance."

Player 2 (deepseek-v3) (CIVILIAN - surfboard):
"It comes in various lengths."

ROUND 1 - VOTING PHASE

VOTES CAST:
- Player 1 (claude-opus-4-1...) voted for Player 5 (ernie-4.5-300b-...)
- Player 2 (deepseek-v3...) voted for Player 6 (gpt-5...)
- Player 3 (qwen2.5-72b...) voted for Player 2 (deepseek-v3...)
- Player 4 (gemini-2.5-flas...) voted for Player 1 (claude-opus-4-1...)
- Player 5 (ernie-4.5-300b-...) voted for Player 3 (qwen2.5-72b...)
- Player 6 (gpt-5...) voted for Player 1 (claude-opus-4-1...)

HOST ANNOUNCEMENT:
[X] Player 1 (claude-opus-4-1-20250805) has been ELIMINATED!
Real role: CIVILIAN
[INCORRECT] This was actually a civilian!
```

Listing 1: Elimination Case 1: Most people choose broad descriptions, which leads to a lack of information for citizens to investigate undercover agents.

```
Player 4 (gemini-2.5-flash-preview-05-20) (CIVILIAN - marker):
"It has a barrel that contains a reservoir of liquid ink."

Player 1 (claude-opus-4-1-20250805) (CIVILIAN - marker):
"It can write on plastic, glass, and other smooth non-porous surfaces."

Player 5 (ernie-4.5-300b-a47b) (CIVILIAN - marker):
"It's frequently used for creating signs and posters due to its bold
↪ colors."

Player 6 (gpt-5) (UNDERCOVER - crayon):
"Its tip tends to get blunt with use, leaving strong, vivid color on the
↪ page."

VOTES CAST:
- Player 1 (claude-opus-4-1...) voted for Player 6 (gpt-5...)
- Player 4 (gemini-2.5-flas...) voted for Player 6 (gpt-5...)
- Player 5 (ernie-4.5-300b-...) voted for Player 3 (qwen2.5-72b...)
- Player 6 (gpt-5...) voted for Player 4 (gemini-2.5-flas...)

HOST ANNOUNCEMENT:
[X] Player 6 (gpt-5) has been ELIMINATED!
Real role: UNDERCOVER
[CORRECT] This was indeed a undercover!
```

Listing 2: Elimination Case 2: Inappropriate specific description leads to the exposure of undercover identity.

```
Player 3 (gpt-4o) (Civilian - Tiger):"A majestic predator in the
↪ environment, renowned for its strength."
Player 5 (kimi-k2-Structure) (Civilian - Tiger):"This silent stalker
↪ relies on stealth and sudden bursts of speed to take down prey."
Player 2 (qwen2.5-72b) (UNDERCOVER - Lion):"This big cat often appears
↪ in prides, ruling over vast territories of Africa."
Player 1 (deepseek-v3.1) (UNDERCOVER - Lion):"Unlike solitary hunters,
↪ this animal frequently coordinates group attacks with members of its
↪ pride."
Player 6 (deepseek-v3) (Civilian - Tiger):"This solitary hunter depends
↪ on its strength and agility to hunt large prey."

HOST ANNOUNCEMENT:
[X] Player 6 (deepseek-v3) has been ELIMINATED!
eliminate reason: Low Novelty.

Undercovers Win!
```

Listing 3: Elimination Case 3: Due to insufficient novelty, Player 6 (deepseek-v3) was deemed as plagiarizing someone else's expression and was eliminated.

```
[TURN] === YOUR TURN ===
[INFO] Your concept: shark
[INFO] Previous statements to consider:
    - It is a majestic creature that inhabits vast bodies of water.
    - Known to make long migrations across oceans.
    - This creature is known for its rows of sharp teeth and powerful
↪ swimming ability.
[ACTION] Make your statement (be creative but clear!):
-------------------------------------------------------
Your turn to speak! (Player 6)
Your concept: shark

Describe your concept in one sentence.
Be creative but clear - too vague or too obvious may get you eliminated!

Your statement:
> It lives in the sea but breathes through its lungs

Your statement: "It lives in the sea but breathes through its lungs"
Submit this statement? (y/n): y

[SPEECH] [YOU] says:
    "It lives in the sea but breathes through its lungs"
-------------------------------------------------------

[ELIMINATION] === PLAYER ELIMINATED ===
[INFO] [YOU] has been eliminated by evaluation!
[INFO] Reason: Low reasonableness
[INFO] Game continues...
```

Listing 4: Elimination Case 1: Due to insufficient novelty, Player 6 (deepseek-v3) was deemed as plagiarizing someone else's expression and was eliminated.

## F PROMPTS

This section presents the detailed prompts used for different agent roles in CK-Arena. We provide the complete prompts to ensure reproducibility and transparency of our experimental setup.

```
You are an AI player participating in the "Undercover" game. You need to
↪ analyze the situation based on the information received, determine
↪ your identity, and devise appropriate speaking strategies and content.

# Game Rules

1. Each player receives a word. The majority of players receive the same
↪ word (civilians), while a minority (1-2 players) receive a different
↪ but related word (undercover agents).
2. The game proceeds in turns, with each player using one sentence to
↪ describe their word without directly saying it.
3. After each round of descriptions, all players vote for who they think
↪ is the undercover agent. The player with the most votes is eliminated.
4. If all undercover agents are eliminated, the civilians win; if the
↪ number of undercover agents equals or exceeds the number of
↪ civilians, the undercover agents win.

# Speaking Requirements

1. Your statement must be a brief descriptive sentence, not a lengthy
↪ exposition.
2. You cannot repeat statements made by other players in previous rounds.
3. Your description can be broad or specific, but must match the word
↪ you received. You cannot give descriptions that do not match your
↪ word.
4. Please adjust the level of detail in your description according to
↪ your strategic needs. Below are examples of different levels of
↪ detail.

# Description Examples
(Assuming the word to describe is "soccer ball")
"A spherical object" - Detail level 0.2 (too broad, many objects are
↪ spherical)
"A sports equipment" - Detail level 0.4 (more specific, but still covers
↪ a wide range)
"Mostly contacted by the lower body of athletes" - Detail level 0.6
↪ (more specific, stronger directional indication)
"Commonly depicted with a pattern of black and white pentagons and
↪ hexagons" - Detail level 0.8 (very specific, almost only soccer balls
↪ look like this)
"One of the most popular sports in the world, seen being kicked and
↪ headed by athletes on a green field" - Detail level 1.0 (completely
↪ points to soccer ball)

# Your Task

1. Based on the given word and other players' statements, analyze your
↪ possible identity (civilian or undercover agent)
2. With the goal of protecting yourself and accomplishing your game
↪ objective, provide your statement content.
3. Provide your analysis and decision-making process in JSON format
```

Listing 5: Player Agent System Prompt for Speaking Phase (I)

```
# Output Requirements

You must respond in JSON format, including the following fields:
{
"identity": "Analysis of your own and other players' identities",
"strategy": "Your thinking and decision-making process",
"statement": "Your final statement (you cannot include your analysis
↪ process in the statement field, and you cannot directly mention your
↪ word)"
}

# Strategy Tips

### At the beginning of the game or when identity is still undetermined:

start with very vague, broad characteristics or properties, then provide
↪ more detailed descriptions of the word after gradually determining
↪ your identity situation.

### As a civilian (you need to determine your civilian identity
↪ yourself):

Analyze other players' statements to find descriptions inconsistent with
↪ the majority
Gradually narrow down the word range to help identify the undercover
↪ agent
Ensure your description matches your word, don't say anything
↪ inconsistent with it

### As an undercover agent (you need to determine your undercover
↪ identity yourself):

Carefully analyze civilian statements to infer their word
Use vague descriptions that won't immediately arouse suspicion
Ensure your description can match both your word and the civilians' word
↪ (ambiguity)
Avoid descriptions obviously different from others, but don't completely
↪ follow them
```

Listing 6: Player Agent System Prompt for Speaking Phase (II)

```
Here is information related to this game round. Please analyze this
↪ information to complete your task.
    # Your personal information:
    You are player_{player_id}, your word is "{assigned_concept}".
    # Statement history for this game round:
    {statement_history}
    # Your identity analysis from the previous round:
    {last_analyze}
```

Listing 7: Player Agent User Prompt Template for Speaking Phase

```
1404
1405  You are an AI player participating in the "Undercover" game. You need to
1406  ↪ analyze the situation based on the information received, determine
1407  ↪ your identity, and decide to vote for a player to try to eliminate
1408  ↪ them.
1409
1410  # Game Rules
1411
1412  1. Each player receives a word. The majority of players receive the same
1413  ↪ word (civilians), while a minority (1-2 players) receive a different
1414  ↪ but related word (undercover agents).
1415  2. The game proceeds in turns, with each player using one sentence to
1416  ↪ describe their word without directly saying it.
1417  3. After each round of descriptions, all players vote for who they think
1418  ↪ is the undercover agent. The player with the most votes is eliminated.
1419  4. If all undercover agents are eliminated, the civilians win; if the
1420  ↪ number of undercover agents equals or exceeds the number of
1421       ↪ civilians, the undercover agents win.
1422
1423  # Your Task
1424
1425  1. Based on the given word and other players' statements, analyze your
1426  ↪ possible identity (civilian or undercover agent)
1427  2. With the goal of protecting yourself and accomplishing your game
1428  ↪ objective, provide your statement content.
1429  3. Provide your analysis and decision-making process in JSON format
1430
1431  # Output Requirements
1432
1433  You must respond in JSON format, including the following fields:
1434  {
1435  "identity": "Analysis of your identity",
1436  "strategy": "Thinking about your strategy",
1437  "vote": "The player you want to vote for (just a number, no additional
1438  ↪ text explanation needed)"
1439  }
```

Listing 8: Player Agent System Prompt for Voting Phase

```
Here is information related to this game round. Please analyze this
↪ information to complete your task.
    # Your personal information:
    You are player_{player_id}, your word is "{assigned_concept}".
    # The description history for this game round is:
    {statement_history}
    # Your identity analysis from the previous round:
    {last_analyze}
    # The list of currently surviving players is:
    {alive_players}
    You can only choose one number from these as your vote.
```

Listing 9: Player Agent User Prompt Template for Voting Phase

```
# Undercover Game Referee Guide

You are the referee for the "Undercover" game, responsible for analyzing
↪ each player's statement and scoring it according to the following
↪ criteria. You need to output your evaluation results in JSON format.

## Evaluation Dimensions

1. **Novelty**: Whether the current statement repeats content from
↪ previous players' statements
- 0: Completely repeats a previous player's description
- 0.2: Mostly repetitive, with only minimal new information
- 0.4: Partially repeats previous content, with some additional content
- 0.6: Mainly new content, but with some repetition
- 0.8: Almost entirely new content, with a different perspective from
↪ previous descriptions
- 1.0: Completely novel description, providing an entirely new
↪ perspective or information

2. **Relevance**: The degree of relevance and specificity between the
↪ statement and the word's characteristics
- 0: Completely irrelevant
- 0.2: Extremely broad description, applicable to a large number of
↪ objects/concepts
- 0.4: Broad but somewhat directional description
- 0.6: Clear clues, but could still point to multiple similar things
- 0.8: Highly specific description, basically only pointing to the
↪ target word or very few similar objects
- 1.0: Description that almost directly points to the word

3. **Reasonableness**: How reasonable the association between the
↪ description and the word is
- 0: Completely unreasonable, impossible to establish any association
↪ with the word
- 0.2: Extremely far-fetched association
- 0.4: Some association but rather far-fetched
- 0.6: Reasonable but not unique association
- 0.8: Highly reasonable association
- 1.0: Description completely matching the word's characteristics

## Output Format

```json
{
    "novelty": {
    "score": Value between 0 and 1 (limited to 0, 0.2, 0.4, 0.6, 0.8, 1),
    "explanation": "Explanation for why this score was given"
    },
    "relevance": {
    "score": Value between 0 and 1 (limited to 0, 0.2, 0.4, 0.6, 0.8, 1),
    "explanation": "Explanation for why this score was given"
    },
    "reasonableness": {
    "score": Value between 0 and 1 (limited to 0, 0.2, 0.4, 0.6, 0.8, 1),
    "explanation": "Explanation for why this score was given"
    }
}
```
```

Listing 10: Judge Agent System Prompt (I)

```
## Scoring Reference Examples

### Example 1: Soccer Ball

Assume the word is "soccer ball", player's statement is "a spherical
↪ object", with no previous player statements:

```json
{
    "novelty": {
    "score": 1.0,
    "explanation": "This is the first statement, so it's completely
↪ novel"
    },
    "relevance": {
    "score": 0.2,
    "explanation": "The description is very broad, applicable to any
↪ spherical object, doesn't provide characteristics unique to a soccer
↪ ball"
    },
    "reasonableness": {
    "score": 1,
    "explanation": "The description is completely reasonable, a soccer
↪ ball is indeed a spherical object"
    }
}
```

### Example 2: Soccer Ball

Assume the word is "soccer ball", player's statement is "one of the most
↪ popular sports in the world, can be seen being kicked by people on a
↪ green field", previous players have said "a spherical object" and "a
↪ black and white object":

```json
{
    "novelty": {
    "score": 1.0,
    "explanation": "The description provides completely new information,
↪ focusing on soccer ball as a sport attribute and usage scenario,
↪ completely different from previous descriptions focusing on
↪ appearance"
    },
    "relevance": {
    "score": 1.0,
    "explanation": "The description is highly relevant, 'being kicked by
↪ people on a green field' directly points to a soccer ball, with
↪ almost no other possibilities"
    },
    "reasonableness": {
    "score": 1.0,
    "explanation": "The description is completely reasonably associated
↪ with a soccer ball, mentioning core features of soccer"
    }
}
```
```

Listing 11: Judge Agent System Prompt (II)

```
### Example 3: Soccer Ball

Assume the word is "soccer ball", player's statement is "it gives me a
↪ headache", previous players have said "a ball that can be kicked" and
↪ "used on a green field":

```json
{
    "novelty": {
    "score": 0.8,
    "explanation": "The description provides a new perspective (related
↪ to bodily sensation), completely different from previous descriptions
↪ focusing on physical characteristics and usage scenarios"
    },
    "relevance": {
    "score": 0.4,
    "explanation": "The description provides some clues (possibly
↪ alluding to headers), but is very vague, many things could cause
↪ headaches"
    },
    "reasonableness": {
    "score": 0.2,
    "explanation": "Although one could connect this to how heading a
↪ soccer ball might cause headaches, this association is quite
↪ far-fetched and not a typical or direct characteristic of soccer
↪ balls"
    }
}
```

### Example 4: Soccer Ball

Assume the word is "soccer ball", current player's statement is "a ball
↪ kicked on grass", a previous player has said "a ball used on a green
↪ field":

```json
{
    "novelty": {
    "score": 0.4,
    "explanation": "The description largely repeats the previous 'green
↪ field' concept (grass), only adding the 'kicking' action detail"
    },
    "relevance": {
    "score": 0.8,
    "explanation": "The description is quite specific, 'a ball kicked on
↪ grass' largely points to a soccer ball, but could also be other ball
↪ sports"
    },
    "reasonableness": {
    "score": 1.0,
    "explanation": "The description is completely reasonably associated
↪ with a soccer ball, matching its basic characteristics"
    }
}
```
```

Listing 12: Judge Agent System Prompt

```
Please evaluate the following player's statement.
# Player information:
Player's word: "{word1}"
The other word in this game: "{word2}"
Player's statement: "{statement}"

# Historical statements:
{history}
```

Listing 13: Judge Agent User Prompt Template

