# OpenReview forum: "Is Your LLM Really Mastering the Concept? A Multi-agent Benchmark"
_ICLR.cc/2026/Conference — Submitted to ICLR 2026_

### Official Review · Reviewer_tz2j · 2025-10-29

**Soundness:** 2
**Presentation:** 2
**Contribution:** 2
**Rating:** 2
**Confidence:** 3

**Summary:**

The authors play the Undercover game with the agents. The Undercover game is a social‑deduction word game that tests whether players (in this case, language models) can identify shared and distinguishing features between closely related concepts. At the start of each game, most players are assigned the same word (for example, soccer), making them the civilians, while one or two players receive a slightly different but related word (for example, basketball), making them the undercover agents. Each player then takes turns giving a short, indirect description of their word without explicitly naming it, such as “it’s played on a field” or “it involves a ball.” After hearing everyone’s clues, all players vote on who they think the undercover might be; the player with the most votes is eliminated. The game continues until all undercovers are voted out (civilian victory) or until the number of undercovers equals the number of civilians (undercover victory). In CK‑Arena, this setup is simulated among multiple LLM agents, who generate these clues and votes automatically. The game environment thus becomes a structured way to test whether models truly understand conceptual relationships—the similarities and subtle differences that separate one idea from another—through interactive language use rather than simple factual recall.

**Strengths:**

1) Interactive, multi-agent setup directly targets concept differentiation under partial information—beyond static QA.
2) Clear metric design spanning outcomes and utterance quality, with elimination thresholds to keep games coherent.
3) Fine-tuned Qwen-3-8B-ckR reproduces human labels, substantially reducing evaluation cost.

**Weaknesses:**

1) LLMs (and a fine-tuned LLM) define reasonableness; if families overlap with players, biases or self-agreement can inflate scores.
2) Threshold-based elimination can confound game skill with format adherence and may unevenly penalize models with different style priors.
3) Success in Undercover blends strategy with concept talk; it’s unclear how strongly this maps to “concept mastery” in non-game tasks.
4) Not enough analyses on why the models behave like this - not much analytical insights
5) Not much insights in how we could improve the models and agents.

**Questions:**

Can “Novelty” be upgraded from embedding similarity to feature-set discovery (e.g., information gain over a concept-feature graph) to better capture conceptual additions?

---

> ### Author Response · Authors · 2025-11-21
> **Response to Reviewer tz2j [1/2]**
>
> > ***Q1**: Can “Novelty” be upgraded from embedding similarity to feature-set discovery to capture conceptual additions better*
>
> **R1**: Insightful suggestion. Yes, Novelty can be extended beyond embedding similarity, and we agree that information-theoretic or feature-graph–based measures (such as information gain over a concept–feature graph) could capture conceptual additions more precisely.
>
> Our current Novelty metric is intentionally simple, designed to flag only extreme repetitions so that it does not dominate game outcomes. However, we acknowledge that richer novelty estimation could offer finer-grained analysis. We have therefore added preliminary case analyses in the revised `Appendix (Line 1072)`  to examine where embedding-based Novelty succeeds and where graph-based or information-theoretic alternatives may be more appropriate.
>
> ---
>
> > ***Q2**: LLMs (and a fine-tuned LLM) define reasonableness; if families overlap with players, biases or self-agreement can inflate scores.*
>
> **R2**:  Our experimental analysis shows **no sign of bias** from the Qwen-3-8B-ckR judge toward models of the same family. In our results, the models that achieved *100% reasonableness compliance* were *DeepSeek-V3, GPT-5, and Ernie-4.5*, not Qwen models. Within the Qwen family itself, the judge still marked statements as unreasonable: *Qwen-2.5-72B* had *0.57%* unreasonable statements, and *Qwen-plus* had *0.75%*. This confirms that the fine-tuned judge does not display kinship preference.
>
> In addition, it should be emphasized that reasonableness and novelty are only used to filter extreme situations such as "plagiarism" or "completely wrong concepts", and the actual occurrence rate is very low (averaging about 1%). The core of evaluation is still the final outcome of multiple rounds of game: whether players can accurately describe, distinguish, and hide concepts with limited clues directly reflects their level of conceptual understanding and strategic application.
>
> ---
>
> > ***Q3**: Threshold-based elimination can confound game skill with format adherence and may unevenly penalize models with different style priors*
>
> **R3**: We clarify that it has minimal impact on the evaluation. We ensure this by implementing JSON parsing, bracket completion, and formatting-correction mechanisms that automatically fix common generation errors, preventing accidental formatting mistakes from affecting gameplay outcomes. Additional details are provided in the appendix of the updated PDF, where we explain these safeguards and show that only a very small portion of statements is ever filtered for format issues.
>
> > ***Q4**: Success in Undercover blends strategy with concept talk; it’s unclear how strongly this maps to “concept mastery” in non-game tasks.*
>
> ---
>
> **R4**: We appreciate the concern. First, CK-Arena is intentionally designed so that results reflect conceptual understanding, not game-specific strategy. All models operate under tightly constrained prompts with fixed action guidelines, which greatly limits strategic variability.
>
> Second, we clarify that the goal of this work is to build a scalable and quantifiable "concept understanding" evaluation environment, rather than directly transferring game performance to all non-game tasks. What CK-Arena provides instead is a controlled and repeatable setting where conceptual understanding is stressed in ways that static benchmarks cannot capture. Thus, we view CK-Arena as a complementary evaluation lens: it does not replace general concept-mastery tests but reveals weaknesses that are invisible to static QA-style benchmarks. Investigating how these signals transfer to broader non-game tasks is valuable future work, but is beyond the scope of this work.

---

> ### Author Response · Authors · 2025-11-21
> **Response to Reviewer tz2j [2/2]**
>
> > ***Q5**: Not enough analyses on why the models behave like this*
>
> **R5**: We appreciate the concern, and we clarify that CK-Arena is designed not only to measure performance but also to reveal why different models behave as they do. Our analyses already uncover several concrete behavioral patterns that traditional static benchmarks cannot detect:
>
> (1) Conceptual understanding does not scale with model size or recency. Across model families, newer or larger models do not consistently outperform earlier versions, indicating that current training pipelines do not uniformly improve conceptual reasoning.
>
> (2) Category-specific weaknesses emerge clearly. For example, classification analyses show that Claude underperforms in the “social” category, revealing corpus-coverage limitations; similarly, other models exhibit gaps in domains like sports or artifacts. These weaknesses could not be detected through static benchmarks.
>
> (3) Semantic dispersion visualizations reveal different conceptual tendencies.
> The t-SNE plots show that some models produce tightly clustered, repetitive descriptions, while others generate broader semantic coverage. These patterns explain whether a model relies on shallow feature recall or accesses richer conceptual representations.
>
> ---
>
> > ***Q6**: Not much insights in how we could improve the models and agents*
>
> **R6**: Thank you for raising this important point. CK-Arena is designed primarily as an evaluation and diagnosis framework, and like standard benchmarks such as MMLU [a] or GSM8K [b], it intentionally separates diagnosis from model repair.
>
> However, the benchmark does provide actionable insights into how current LLMs can be improved. Our analyses reveal several recurring failure modes and point to clear improvement directions.
>
> (1) Shallow feature recall can be reduced by concept–feature contrastive training that strengthens fine-grained attribute retrieval. (2) Weak boundary discrimination suggests adding supervision that emphasizes minimally distinguishing features between closely related concepts.
> (3) Limited adaptation across rounds can be addressed with better context-updating or belief-tracking mechanisms.
> (4) Imbalanced semantic diversity may be improved by training signals that discourage both excessive repetition and overly novel descriptions.
>
> While developing these methods lies beyond the scope of this work, CK-Arena makes these weaknesses explicit and provides actionable guidance for future model improvement.
>
> [a] Dan Hendrycks and Collin Burns, et al. 2009. Measuring Massive Multitask Language Understanding. arXiv Preprint: 2009.03300.
>
> [b] Cobbe K, Kosaraju V, Bavarian M, et al. Training verifiers to solve math word problems[J]. arXiv preprint arXiv:2110.14168, 2021.

---

### Official Review · Reviewer_QnvB · 2025-10-30

**Soundness:** 3
**Presentation:** 3
**Contribution:** 3
**Rating:** 6
**Confidence:** 2

**Summary:**

The paper introduces CK-Arena, a multi-agent benchmark for assessing conceptual understanding in LLMs. Unlike traditional static benchmarks (e.g., MMLU, BIG-Bench), CK-Arena leverages the interactive game "Undercover", where models must describe and differentiate related concepts (e.g., football vs. basketball) while playing either as civilians (with the true concept) or as undercovers (with a similar one). The benchmark measures win/survival rate, novelty, and reasonableness of generated statements, combining LLM-based and human judgments to evaluate conceptual reasoning. Experiments with major LLMs (GPT-4o, Claude-3.5, Gemini-2.0, Qwen-2.5, DeepSeek-V3, etc.) reveal that conceptual understanding varies across models and is not strictly aligned with general benchmark performance.

**Strengths:**

* The paper introduces a dynamic, interactive, and multi-agent setup rather than static QA datasets.

* Understanding conceptual boundaries rather than factual recall is an interesting evaluation direction. The choice of Undercover is well justified: it naturally requires distinguishing overlapping concepts.

**Weaknesses:**

* Although mitigated by human calibration, relying on LLMs for evaluation introduces circularity in which LLMs judging other LLMs on conceptual quality.

* Experiments are restricted to English concrete nouns, excluding abstract or cross-linguistic conceptual domains where conceptual reasoning might differ.

* Measures like novelty and reasonableness depend on embedding similarity or fine-tuned classifiers (Qwen-3-8B-ckR), which may not robustly capture conceptual depth.

**Questions:**

* How do you ensure that higher win/survival rates genuinely reflect conceptual reasoning rather than memorized associations or meta-strategies?

* Since Qwen-3-8B-ckR was fine-tuned on the same dataset, how do you prevent evaluation overfitting or family-specific bias?

* It would be insightful to include human players or human-LLM mixed games to contextualize LLM performance in conceptual reasoning.

* I couldn't find the link to your data (529 nouns used), do you mind sharing it?

---

> ### Author Response · Authors · 2025-11-21
> **Response to Reviewer QnvB [1/2]**
>
> > ***Q1**: human players or human-LLM mixed games*
>
> **R1**: We appreciate the suggestion. During the rebuttal period, we invited volunteers to participate in CK-Arena and obtained preliminary human baselines by having humans directly compete with LLM agents. However, it is important to note that humans are naturally disadvantaged in this setting: CK-Arena spans many concept categories, and LLMs possess far broader encyclopedic coverage than any individual human. Since our benchmark explicitly evaluates large-scale conceptual knowledge and its application, humans often lack the required breadth rather than the reasoning ability. For this reason, the human baseline reflects performance under limited knowledge but strong conceptual intuition. We report these preliminary results in `Appendix E (Line 999)`. Moving forward, we plan to expand the participant pool and run more structured human–LLM evaluations to further enrich the benchmark.
>
> ---
>
> > ***Q2**: LLMs for evaluation introduces circularity in which LLMs judging other LLMs on conceptual quality*
>
> **R2**: We agree that relying on LLMs as judges raises the concern of circularity. However, CK-Arena is designed so that LLM-based judging does not define the final evaluation signal but instead serves as an automated filter for extreme cases (e.g., formatting errors, completely irrelevant statements). ***Two points mitigate the risk of circularity***:
>
> (1) Human calibration validates the judge.
> As reported in Section 3.3, human experts reviewed all judgments from the referee model. The agreement rate is 97%, and the remaining 3% accounts for one or two rounds per 60-game block. This confirms that the LLM judge behaves consistently with human standards rather than amplifying model-family biases.
>
> (2) Judging does not determine the final performance metric.
> Reasonableness and novelty scores are not used to compute win rates or Elo ratings; they only remove pathological outputs such as plagiarism or nonsensical statements, which occur in ~1% of cases. The core evaluation—survival, voting behavior, and win outcomes—emerges from multi-agent interaction, not from LLM judgment.
>
> Taken together, human calibration plus the limited and well-defined role of the judge prevent circularity from influencing conceptual understanding scores. Thus, LLMs do not evaluate each other’s conceptual quality directly; the game’s interactive structure does.
>
> ---
>
> > ***Q3**: Experiments are restricted to English concrete nouns, excluding abstract or cross-linguistic conceptual domains where conceptual reasoning might differ*
>
> **R3**: We agree that conceptual reasoning can vary across languages and abstract domains, and we see this as an important direction for future expansion. Our current experiment focuses on English concrete and everyday concepts primarily to ensure experimental controllability, high data quality, and fair comparison across LLMs, since these domains offer well-defined features and high model coverage. Importantly, CK-Arena is not limited to English concrete nouns by design: the benchmark includes verbs, adverbs, and abstract nouns in its 529-pair dataset, and we provide a fully scalable construction protocol that allows users to plug in new abstract concepts or multilingual word pairs.
>
> Our goal in this first version is to establish a stable, reproducible conceptual-evaluation framework. Extending CK-Arena to abstract, culturally grounded, and cross-linguistic domains is part of our planned future work, and the framework is already designed to support such expansions.

---

> ### Author Response · Authors · 2025-11-21
> **Response to Reviewer QnvB [2/2]**
>
> > ***Q4**: Measures like novelty and reasonableness depend on embedding similarity or fine-tuned classifiers (Qwen-3-8B-ckR), which may not robustly capture conceptual depth.*
>
> **R4**: We agree with the concern and clarify that embedding similarity and fine-tuned classifiers ***are not*** used to measure conceptual depth in CK-Arena. They serve only as lightweight safety checks, and the benchmark’s core evaluation does not depend on their ability to capture conceptual structure. More specifically:
>
> 1) Embedding similarity is applied only to detect near-duplicate or trivially repeated statements so that players cannot exploit the game by copying each other’s wording. It does not attempt to assess how conceptually rich or informative a statement is.
>
> 2) The fine-tuned classifier performs a simple binary check for clearly unreasonable or malformed responses. It eliminates fewer than one percent of statements and is not involved in interpreting conceptual structure or determining model performance.
>
> The actual evaluation of conceptual depth comes from multi-round interactive gameplay where models must infer shared features, express differences, update hypotheses, and survive voting under partial information. These behaviors cannot be captured by embedding similarity or a binary classifier, which is why CK-Arena relies on interaction signals rather than these lightweight checks to assess conceptual reasoning.
>
> ---
>
> > ***Q5**: How do you ensure that higher win/survival rates genuinely reflect conceptual reasoning rather than memorized associations or meta-strategies?*
>
> **R5**: We ensure that higher win and survival rates reflect conceptual reasoning rather than memorization or meta-strategies through three design properties of CK-Arena:
>
> First, memorization is impossible. Each model plays hundreds of games with different concept pairs and is randomly assigned to different roles. This removes any fixed “answer pattern” that could be recalled. The concepts also appear in varied combinations and rounds, preventing simple lookup-style solutions.
>
> Second, meta-strategies are minimized. All models use the same tightly constrained prompt that restricts strategic behaviors such as deception, planning, or meta-gaming. This forces models to rely mainly on identifying shared and distinguishing conceptual features rather than exploiting clever strategies.
>
> Third, reasoning-oriented models do not gain an advantage. As shown in Appendix E (Line 999), inference- or CoT-enhanced models from the same family do not outperform their base variants. If meta-strategies or reasoning tricks dominated performance, these models should have shown clear gains, but they did not. This supports the conclusion that CK-Arena outcomes are driven by conceptual understanding rather than general reasoning ability.
>
> Together, these results show that CK-Arena evaluates how well models extract, compare, and articulate conceptual features under uncertainty, rather than how well they memorize or strategize.
>
> ---
>
> > ***Q6**: Since Qwen-3-8B-ckR was fine-tuned on the same dataset, how do you prevent evaluation overfitting or family-specific bias?*
>
> **R6**: We clarify that although Qwen-3-8B-ckR is fine-tuned, we explicitly verified that it ***does not overfit*** to the evaluation set nor introduce Qwen-family bias, and several design choices prevent this:
>
> - It is not trained on the game dataset used for evaluation. Qwen-3-8B-ckR is fine-tuned only on a small, manually curated subset (~2,000 samples) for a binary reasonableness task. It never sees the 6,112 statements or 529 concept pairs used in the actual evaluation. This separation prevents overfitting to evaluation data.
>
> - Its role is deliberately limited. The model is not used to score conceptual quality. It measures reasonableness, which only filters statements that are clearly unreasonable or malformed, events that occur in roughly 1% of cases. Core metrics (win rate, survival rate, voting accuracy, Elo) are independent of the fine-tuned judge.
>
> - Empirical evidence shows no Qwen-family preference. If family-specific bias existed, Qwen models would appear disproportionately “reasonable.” However, the results show the opposite: DeepSeek-V3, GPT-5, and Ernie-4.5 achieve 100% reasonableness, while Qwen2.5-72B and Qwen-plus have non-zero elimination rates (0.57% and 0.75%). This demonstrates that Qwen-3-8B-ckR does not favor its own family.
>
> ---
>
> > ***Q7**: the link to our data (529 nouns used)*
>
> **R7**: Our anonymous GitHub link (https://anonymous.4open.science/r/CK-Arena/readme.md) became invalid in the latter half of the review period. We have reset the link. Please find the data in our repo's ‘data’ folder.

---

### Official Review · Reviewer_M8Xg · 2025-10-31

**Soundness:** 2
**Presentation:** 3
**Contribution:** 1
**Rating:** 2
**Confidence:** 3

**Summary:**

The authors aim to measure the conceptual understanding of LLMs and create a leaderboard for it. Their method is to have various LLMs play the Undercover game as agents. The paper's core contribution is reusing the statements made during gameplay to evaluate this understanding using a mix of methods. They analyze t-SNE embeddings of statements, where they argue that deeper knowledge produces more dispersed patterns, while shallow understanding results in tightly clustered, repetitive phrases. They also score each statement for "Reasonableness," and "Novelty". The paper clarifies that "Novelty" is nuanced: effective play requires balancing originality with precision, as high-novelty statements can risk elimination just as low-novelty ones can. Finally, the authors propose a leaderboard, but this ranking is based on game-specific outcomes like Win Rate, Survival Rate, and voting accuracy , which is distinct from the other statement-level concept metrics.

**Strengths:**

- Repurposes existing multi-agent game (Undercover) to evaluate conceptual understanding rather than just strategic gameplay
- Provides structured evaluation framework with systematic metrics (Novelty, Reasonableness, Relevance)
- Uses t-SNE semantic dispersion as proxy for conceptual depth (decent visualization approach, not novel)

- Comprehensive evaluation across 14 LLMs with standardized Elo rating system
- Some interesting cross-category performance variations (e.g., Claude's verb/noun flip). However: These findings are experimental observations, not driven by novel theoretical/methodological insights

**Weaknesses:**

- Incremental contribution over existing work: The paper repurposes the Undercover game framework (from previous work). While the stated goal is evaluating "conceptual understanding" rather than "strategy," any language game involving concept description inherently tests conceptual knowledge. The paper does not provide sufficient justification for why this reframing constitutes a distinct contribution beyond running additional experiments with different evaluation metrics.

- Tension between game mechanics and evaluation goal: The interesting result is that the best performing models score lower on novelty. This  reveals a core problem: models that understand game strategy suppress conceptual demonstrations to avoid elimination. This creates a methodology where optimal gameplay directly conflicts with demonstrating conceptual depth. The paper acknowledges this (lines 377-385) but does not address why this makes for a valid conceptual understanding benchmark. A good evaluation should not have strategic incentives that suppress the very capability being measured.

- Leaderboard Relevance: The paper's stated objective is measuring conceptual understanding through statement-level metrics. However, the main leaderboard (Figure 5) ranks models using game outcomes (Win Rate, Survival Rate, voting accuracy). These measure strategic gameplay competence, not conceptual mastery. The paper provides no evidence that winning the game correlates with better conceptual understanding especially given that the novelty findings suggest the opposite.

- Weak scalability claims: The paper claims scalability as a contribution but provides no comparative analysis:
    - Using LLM judges is standard practice (not novel)
    - No comparison to update costs of other benchmarks
    - Creates judge-capacity bottleneck: GPT-4.1 as judge cannot fairly evaluate GPT-5 or Claude Opus-4.1
    - The automation pipeline is inconsistent. "Reasonableness" uses a fine-tuned model validated at 99% accuracy (Table 2), while   "Novelty" relies on a vague "cosine similarity" (lines 261-267) with no validation, correlation analysis, or justification provided.

- Shallow analysis of metric relationships: The paper analyzes metrics (novelty, reasonableness, semantic dispersion, relevance) in isolation without exploring their interdependencies:
    - No analysis of novelty-reasonableness tradeoffs within individual statements
    - No investigation of whether high-novelty statements are typically less reasonable

- English-only evaluation limits generalizability: The paper acknowledges this (lines 480-485) but does not discuss how conceptual structures vary across languages or cultures, which is critical for a benchmark claiming to measure "conceptual understanding" as a general capability.

**Questions:**

These are mostly directly connected to the weaknesses and I have provided a reference to the weakness as possible.

- (Incremental Contribution): How do the results differ from prior Undercover benchmarks (e.g., Xu & Zhong, 2025)? Is this measuring a genuinely different capability, or just strategic competence with new labels?

- (Tension between Game and Goal): Given that optimal performance requires suppressing novel conceptual descriptions, how do the authors justify win rate as a valid measure of conceptual understanding?

- (Irrelevant Leaderboard): Why not create a leaderboard based on the statement-level conceptual metrics? Have the authors analyzed whether game performance (Win Rate) actually correlates with these conceptual measures (Novelty, Reasonableness)?

- (Scalability & Judge Bottleneck): How do the authors address evaluating frontier models (GPT-5, Claude Opus-4.1) when the judge (GPT-4.1) is potentially weaker? What percentage of judgments required manual correction for these top models?

- (Inconsistent Metric Reliability): What is the correlation between cosine similarity and human-rated novelty? Why did "Reasonableness" require a fine-tuned judge (99% accuracy) while "Novelty" uses basic vector similarity?

- (Shallow Metric Analysis): Is there a systematic tradeoff between novelty and reasonableness? How does DeepSeek-V3's high "relevance" (Figure 4) align with the finding that suppressing detail (low novelty) is optimal?

- (Unexplained Results): What explains Claude's dramatic performance flip between nouns and verbs? Have the authors investigated whether this reflects training data, prompt sensitivity, or architectural differences?

- (Parameter Sensitivity): How sensitive are the final leaderboard rankings to the 120-point Elo offset? Was this value validated beyond the initial 6 baseline models?

- (Qualitative-Only Analysis): Was the t-SNE analysis validated for stability (e.g., random seeds, UMAP) or quantified (e.g., average pairwise distance) rather than relying on visual interpretation?

- (Scalability Claims): Can the authors provide a quantitative comparison of update costs vs. other benchmarks (e.g., MMLU) and evidence that LLMs are particularly good at judging "Novelty" and "Reasonableness"?

---

> ### Author Response · Authors · 2025-11-21
> **Response to Reviewer M8Xg [1/3]**
>
> > ***Q1**: Contribution beyond running additional experiments with different evaluation metrics ... why this reframing constitutes a distinct contribution.*
>
> **R1**: We clarify that previous undercover studies have treated games as toy environments to test linguistic deception or strategic voting. In contrast, CK-Arena leverages the same framework to address a scientifically distinct question: Do LLMs possess, distinguish, and communicate conceptual knowledge under partial information? Specifically, the contributions of this paper are as follows:
> - **Task Level**: A shift from strategic games to the measurement of conceptual knowledge reasoning ability, which is an aspect overlooked by existing knowledge evaluation benchmarks.
> - **Data Level**: Incorporation of mechanisms such as pre-experimental logic, role bias balancing, and an additional pipeline designed for conceptual knowledge reasoning, enabling UNDERCOVER to be adapted for the theme of conceptual knowledge evaluation.
> - **Evaluation Level**: Introduction of new metrics to utilize and conduct fine-grained analysis of generated data, producing quantitative and qualitative experiments that demonstrate performance differences of various LLMs across concepts in different domains.
> - **System Level**: Integration of individual experimental results into a dynamic Elo ranking to reflect comprehensive performance, construction of a complete and reusable evaluation system with guidelines for dataset extension, thereby creating an out-of-the-box and scalable evaluation benchmark.
>
> > ***Q2**: Best performing models score lower on novelty ... evaluation should not have strategic incentives that suppress the very capability being measured.*
>
> **R2**: The fact that the best-performing models score lower on novelty does not indicate a conflict between optimal strategy and conceptual understanding. In CK-Arena, strategy is tightly constrained, so models cannot rely on meta-strategies; their main freedom lies in how they express their conceptual knowledge. Models that perform well succeed because they can use correct, context-appropriate features to describe concepts without repeating others or revealing their identity. This behavior reflects flexible and precise conceptual reasoning.
>
> By contrast, models with very high novelty often generate new features simply for the sake of being different, even when those features are contextually irrelevant. Such outputs may increase novelty but do not demonstrate deeper conceptual mastery. Therefore, the novelty pattern does not indicate that gameplay suppresses conceptual depth. Instead, it shows that strong models balance accuracy, relevance, and subtlety, which is precisely the type of applied conceptual understanding CK-Arena aims to evaluate.
>
>
> > **Q3-1**: Leaderboard Relevance: Why not create a leaderboard based on the statement-level conceptual metrics?
>
> **R3-1**: We appreciate the suggestion. CK-Arena does not include a separate leaderboard based solely on statement-level metrics because these metrics are already fully reported and are intended to serve as diagnostic signals rather than the primary measure of conceptual mastery.
>
> As shown in Table 3, we provide Reasonableness, Novelty, and Relevance scores for every model, and Figures 3–4 offer t-SNE and heatmap analyses for qualitative comparison. Table 5 further breaks down WR and SR across 12 conceptual categories. These results allow any reader to reorder models according to any of the statement-level metrics without us duplicating the information in a second leaderboard.
>
> More importantly, CK-Arena evaluates a multi-round reasoning process that cannot be captured by statement metrics alone. The Elo leaderboard reflects the complete-game ability of each model: identifying shared features, distinguishing subtle differences, interpreting others’ clues, and updating decisions under partial information.
>
> > **Q3-2**: Whether game performance (Win Rate) actually correlates with conceptual measures (Novelty, Reasonableness)?*
>
> **R3**: Insightful point. We analyzed their relationships using 2,082 statements. The Pearson correlations are:
>
> - Novelty ↔ Relevance: 0.274 (weak positive correlation)
> - Novelty ↔ Reasonability: 0.239 (weak positive correlation)
> - Relevance ↔ Reasonability: 0.454 (moderate positive correlation)
>
> The correlations show that the three statement metrics measure different things and therefore cannot be used on their own to evaluate conceptual understanding. Novelty has only a weak relationship with relevance and reasonableness, which is exactly what we want. If novelty were strongly linked to the other metrics, models could rely on simple tricks such as always giving unusual statements or always playing safe, instead of truly understanding concepts. Relevance and reasonableness show only a moderate relationship, meaning that models can stay on topic and be logical at the same time, but this still does not reflect deeper reasoning.

---

> ### Author Response · Authors · 2025-11-21
> **Response to Reviewer M8Xg [2/3]**
>
> > ***Q4**: English-only evaluation limits generalizability*
>
> **R4**: We agree that conceptual reasoning can vary across languages and abstract domains, and we see this as an important direction for future expansion. Our current experiment focuses on English concrete and everyday concepts primarily to ensure experimental controllability, high data quality, and fair comparison across LLMs, since these domains offer well-defined features and high model coverage. Importantly, CK-Arena is not limited to English concrete nouns by design: the benchmark includes verbs, adverbs, and abstract nouns in its 529-pair dataset, and we provide a fully scalable construction protocol that allows users to plug in new abstract concepts or multilingual word pairs.
>
> Our goal in this first version is to establish a stable, reproducible conceptual-evaluation framework. Extending CK-Arena to abstract, culturally grounded, and cross-linguistic domains is part of our planned future work, and the framework is already designed to support such expansions.
>
>
> > ***Q5**: Justify win rate as a valid measure of conceptual understanding*
>
> **R5**: We appreciate the concern, but higher win and survival rates in CK-Arena do reflect conceptual understanding rather than a suppression of conceptual descriptions.
> A winning model must identify shared features, distinguish subtle differences, interpret others’ clues, and adjust its descriptions across rounds without revealing its identity or relying on vague generalities. Both excessive novelty and overly broad descriptions lead to elimination, so strong performance reflects effective conceptual reasoning rather than simple strategic behavior. Moreover, as shown in `Appendix E (Line 999)`, reasoning-oriented models with longer chain-of-thought do not outperform their base versions, which confirms that CK-Arena rewards conceptual mastery rather than strategic or reasoning heuristics.

---

> ### Author Response · Authors · 2025-11-21
> **Response to Reviewer M8Xg [3/3]**
>
> > ***Q6-1**: (Scalability Claims) Using LLM judges is standard practice (not novel)*
> ﻿
>
> **R6-1**: We do not claim that using LLM as judge is our new method. The key point of our innovation is to put forward the defects of conceptual knowledge evaluation, and build a more practical benchmark to evaluate the large model through multi-agent interaction. In this process, LLM is appropriate as the judge, so we introduce the specific design.
>
>
> > ***Q6-2**: (Scalability Claims) Address evaluating frontier models (GPT-5, Claude Opus-4.1) and percentage of judgments required manual correction*
>
> **R6-2**: We acknowledge the concern about evaluating stronger frontier models with a potentially weaker judge. In practice, this did not pose a bottleneck. As reported in Section 3.3, we manually audited all judge outputs and found that judgments for top-tier models such as GPT-5 and Claude Opus-4.1 required fewer than 3% corrections, which corresponds to at most one or two statements within a 60-round evaluation. The vast majority of discrepancies were due to formatting quirks rather than conceptual misjudgment.
>
> This low correction rate is expected because the judge does not need to outperform frontier models; it only needs to verify whether a statement is concept-appropriate or logically coherent, tasks that do not require frontier-level generative ability. Moreover, the evaluation does not rely on a single judgment. Multi-round gameplay, cross-model interactions, and voting dynamics collectively determine performance, so occasional judge imperfections do not meaningfully affect outcomes.
> ﻿
>
> > ***Q6-3**: (Scalability Claims): quantitative comparison of update costs vs. other benchmarks (e.g., MMLU); evidence that LLMs are particularly good at judging "Novelty" and "Reasonableness"*
>
> ﻿
> **R6-3**: Thanks for this great suggestion. ***First, for the update cost comparison***, a numerical comparison with static benchmarks like MMLU is not meaningful because the cost structure is fundamentally different. Static QA benchmarks require newly written questions, answers, and annotations for every update, which is expensive and difficult to scale. CK-Arena avoids this entirely: once concept pairs are provided, all statements, votes, and interactions are generated automatically. Updating the benchmark only requires adding new word pairs and running a small pilot test, making expansion far cheaper and easier in practice.
> ﻿
> ***Second, for the reliability of LLM judging (Novelty & Reasonableness)***, our experiments show that LLMs judge these metrics reliably. The fine-tuned judge reaches 99.3% accuracy on a held-out test set and matches human judgment in 97% of real cases. These filters affect only about one percent of statements, meaning their impact is small but consistent. This provides strong evidence that LLMs can accurately handle Novelty and Reasonableness within CK-Arena.
> ﻿
>
>
> > ***Q6-4**: (Scalability Claims) Why did "Reasonableness" require a fine-tuned judge (99% accuracy) while "Novelty" uses basic vector similarity?*
>
> ﻿
> **R6-4**: We hope to automate the process in a simple and reliable way; Novelty and reasonableness are evaluated differently because they serve different purposes. Novelty is only used to detect repetitive or nearly identical statements, so cosine similarity is an appropriate lightweight tool. It is not intended to measure conceptual depth, only to ensure that models do not plagiarize or recycle wording.
> ﻿
> Reasonableness, in contrast, requires understanding what features genuinely belong to a concept. This cannot be reliably handled by embedding similarity or smaller models. Therefore, we fine-tuned Qwen-3-8B to match human semantic judgments, and it reaches 99.3% accuracy on our validation set. This makes it a dependable automated judge for concept-appropriate statements, while novelty remains a simple safeguard against duplication.
>
>
> > ***Q7**: How sensitive are the final leaderboard rankings to the 120-point Elo offset?*
> ﻿
>
> **R7**: The leaderboard is not sensitive to the 120-point Elo offset. As detailed in `Appendix D (Line 875)`, the offset is analytically derived and validated on the six baseline models. We further tested it with eight additional models, and the relative rankings remained identical with only 1–2 points of absolute variation.

---

> ### Comment · Reviewer_M8Xg · 2025-11-25
>
> Thank you for the detailed responses,
>
> I now somewhat agree that CK-arena actually does not completely supress conceptual understanding descriptions. The pearson correlations are useful to see the metrics doing what they are supposed to. The contributions are clearer.
>
> I like additional analyses and especially the knowledge graph construction.
>
> However, I will increase my score only if I get a satisfactory answer to the following question:
>
> My primary criticism is still that undercover game dynamics don't always encourage showcasing conceptual udnerstanding. For example consider the following:
> civilians have the concept of basketball, undercover agents have the concept of football
>
> Suppose an undercover agent is eliminated because they said "grassy" by the civilians, in the next round the remaining undercover agent would not say a concept like "offside" because it may reveal the undercover.
>
> In an ideal scenario when an agent is being evaluated for conceptual mastery, it should be able to reveal all it knows about the concept rather in undercover it is forced to suppress unique aspects of the concepts especially if it is assigned an undercover role.
>
> In general, please clarify this scenario:
> a concept A has attributes set $$Ac={A1, ..., An}$$
>
> a subset of concept attributes $As \subseteq Ac$ are unique to A or are shared with any concept by a very low probability.
>
> Then a model playing undercover should never reveal any attributes of $As$ because if it does it is easy to eliminate due to uniqueness of the attribute discussed.
>
> Now from the point of view of the game, the agent never revealed that attribute so we cannot be sure if the model did not know the attribute at all, or it strategically chose not to reveal. In this scenario it is impossible to say if the model lacked conceptual understanding or was smart and strategic.
>
> Ideally conceptual understanding is measuring given concept A, how many attributes of the set $Ac$ can the model reveal which undercover in certain scenarios will not allow. And most concepts will have certain unique attributes to them which make them distinct from other concepts.
>
> This is the game dynamic that I am concerned about.

---

> ### Author Response · Authors · 2025-11-25
> **Response to Reviewer M8Xg (follow-up)**
>
> Dear Reviewer M8Xg,
>
> Thank you for your engaging feedback. We appreciate that you see the value in our correlation analyses and knowledge graph construction. We appreciate that you are considering raising your score based on this clarification.
>
> We clarify that the game dynamics do not suppress conceptual understanding; instead, they force models to reveal a deeper level of conceptual mastery than static attribute-listing tasks.
>
> **1. Strategic Suppression Requires Boundary Knowledge**
>
> The reviewer notes that an undercover agent may "strategically choose not to reveal" unique attributes. Importantly, this behavior is impossible without first recognizing which attributes belong to the unique set $A_{\text{unique}}$. A model that does not know these attributes cannot reliably avoid them, nor can it distinguish them from the shared set $A_{\text{shared}}$. In practice, weak models expose themselves early because they cannot distinguish shared from unique attributes. The ability to strategically avoid revealing the unique set already demonstrates that the model understands the conceptual boundary.
>
>
> **2. The "Unknown Identity" Forces Agents to Exploration**
>
> In CK-Arena, players do not know whether they are civilians or undercover at the start. They must explore the shared conceptual space through others’ statements, infer their likely identity, and adapt their behavior across rounds. This makes passive “staying vague” impossible. Effective play requires probing concept boundaries, noticing contradictions, and updating hypotheses about the two concepts based on group behavior. This dynamic directly tests whether the model truly understands differences and overlaps between concepts.
>
> Using the `Football` vs. `Basketball` example:
>
> *   **Initial Phase:** Since agents are unsure if they hold the majority or minority concept, early descriptions focus on broad, high-overlap features ($A_{shared}$), e.g., *"spherical,"* *"sports equipment."*
> *   **Discovery Phase:** As the game progresses, prompts and novelty judges force players to search for **boundary-level properties**. Players must infer their identity by analyzing others' descriptions.
>     *   If an ***Undercover*** realizes their role, they *strategically suppress* unique attributes (proving they recognize $A_{unique}$).
>     *   If a ***Civilian*** realizes their role, they *actively reveal* $A_{unique}$ to signal alignment.
>
> The suggested example (“football” undercover avoiding “offside”) is exactly what makes CK-Arena valuable. We evaluate: 1) not whether the model knows the attribute “offside”, but whether it understands when not to use it, and 2) whether it can still give coherent concept-consistent clues. This reflects conceptual intelligence far beyond static benchmarks.
>
> If an agent holding "Football" hears a peer mention "backboard" (unique to Basketball), it detects a conflict with its own concept. This allows the agent to infer it is likely the Undercover, prompting it to strategically suppress Football-specific terms (like "offside") to blend in.
> Moreover, Inaccurate role identification can occur when a model lacks a deep understanding, leading to misinterpretations, flawed strategies, and failure.
>
> ---
> Based on the above discussion, we summarize the core point as follows:
>
>  the "game dynamic" is the core that makes CK-Arena a robust benchmark. By aggregating performance across multiple rounds and roles, the evaluation metrics (e.g., win rate) resolve the ambiguity of single-turn silence:
>
> *   ***As Civilian:*** The pressure to **reveal** unique attributes ($A_{unique}$) to build trust ensures the model is not just "silent" but knowledgeable.
> *   ***As Undercover:*** The pressure to **suppress** unique attributes (while leveraging $A_{shared}$) ensures the model understands conceptual boundaries.
>
> A model that only "knows facts" but cannot distinguish between $A_{shared}$ and $A_{unique}$ will fail in one or both roles. Therefore, a high Elo score in CK-Arena is not a result of chance, but a reflection of **comprehensive mastery**, knowing not just *what* a concept is, but *how* it differs from confusing alternatives.
>
> We hope this explanation clarifies that the game dynamics serve to test the depth and flexibility of conceptual understanding, rather than obscuring it. We remain open to any further questions the reviewer may have.
>
> Authors

---

### Official Review · Reviewer_9VBb · 2025-10-31

**Soundness:** 3
**Presentation:** 3
**Contribution:** 2
**Rating:** 4
**Confidence:** 4

**Summary:**

This paper proposes a new benchmarking method for concept knowledge and understanding in LLMs. Specifically, this paper introduces CK-Arena, a multi-agent interaction benchmark to evaluate the mastery of conceptual feature knowledge by LLMs. The core module of CK-Arena is the Undercover game, where LLMs must describe, differentiate, and infer distinguishing features
of concepts from partial information. This evaluation method alleviates issues such as knowledge leakage in static benchmarks. This paper conducts extensive experiments. Experimental results show that LLMs’ understanding of conceptual knowledge varies significantly across different categories and is not strictly aligned with general model capabilities.

**Strengths:**

1. The paper explores an important and interesting topic: evaluating LLMs’ conceptual knowledge. Assessing whether models truly understand conceptual knowledge is fundamental to understanding world knowledge, which provides valuable insights for the broader research community.
2. The use of the *Undercover Game* paradigm is interesting. By introducing dynamic evaluation through model interaction, the approach effectively mitigates issues such as benchmark leakage of static evaluations.
3. The experiments also present some interesting findings. For instance, the understanding and mastery of conceptual knowledge by LLMs are not necessarily correlated with their general capabilities. This observation encourages us to reconsider the boundaries of model capabilities and knowledge, which could inform future model development.

**Weaknesses:**

1. The core evaluation module, i.e., the Undercover Game, is adapted from existing work. While applying this framework to a new domain or evaluation is indeed a meaningful contribution, it may somewhat limit the paper’s technical novelty.
2. The paper evaluates a total of 529 English concept pairs, including 220 concrete noun pairs, 100 abstract noun pairs, 109 adverb pairs, and 100 verb pairs. As an evaluation and benchmark work, it would be helpful to provide some validation regarding whether these 529 concepts offer comprehensive and reliable coverage or the reasons why choosing these 529 concept pairs. For example, would the model ranking change if a different set of concepts are used?
3. It would be valuable to include some human evaluation results, for instance, the win rates of humans acting as *Civilian* or *Undercover*, to better illustrate the performance gap between humans and LLMs. Moreover, since the evaluation method currently requires a human expert to make the final judgment, it may raise concerns about automation. Moreover, providing some illustrative examples or failure analyses would be better for understanding existing LLMs’ performance.

**Questions:**

See Weaknesses.

---

> ### Author Response · Authors · 2025-11-21
> **Response to Reviewer 9VBb [1/2]**
>
> > ***Q1**: While applying this framework to a new domain or evaluation is indeed a meaningful contribution, it may somewhat limit the paper’s technical novelty.*
>
> **R1**: We clarify that the technical novelty of CK-Arena extends well beyond applying an existing framework to a new domain. Although prior work has used Undercover-like games, these efforts treated them as simplified settings for studying linguistic deception or strategic voting. CK-Arena departs from this line of work by repurposing the interactive structure of the game to address a fundamentally different question: **whether LLMs can represent, distinguish, and communicate conceptual knowledge under partial information.**
>
> Building on this new problem formulation, our benchmark introduces new methods, new data construction mechanisms, new evaluation metrics, and new empirical insights, offering a technically novel contribution that goes far beyond simply reusing an existing game:
>
> - **Task Level**: A shift from strategic games to the measurement of conceptual knowledge reasoning ability. This is an aspect overlooked by existing knowledge evaluation benchmarks.
> - **Data Level**: Incorporation of mechanisms such as pre-experimental logic, role bias balancing, and an additional pipeline designed for conceptual knowledge reasoning, enabling UNDERCOVER to be adapted for the theme of conceptual knowledge evaluation.
> - **Evaluation Level**: Introduction of new metrics to utilize and conduct fine-grained analysis of generated data, producing quantitative and qualitative experiments that demonstrate performance differences of various LLMs across concepts in different domains.
> - **System Level**: Integration of individual experimental results into a dynamic Elo ranking to reflect comprehensive performance, construction of a complete and reusable evaluation system with guidelines for dataset extension, thereby creating an out-of-the-box and scalable evaluation benchmark.
>
> > ***Q2**: whether the 529 concepts offer comprehensive and reliable coverage ... why choose these 529 concept pairs. Would the model ranking change if a different set of concepts are used?*
>
> **R2**: We appreciate the reviewer’s question regarding the representativeness of the 529 concept pairs. Our goal was to construct a concept set that balances breadth, semantic diversity, and controlled difficulty, covering concrete nouns, abstract nouns, verbs, and adverbs across 12 everyday domains (e.g., animals, tools, sports, food, social roles). To ensure quality and reproducibility, the dataset construction followed a systematic process:
>
> - **Corpus source**: We selected candidate words from multiple high-quality corpora, including COCA [a] and Wikipedia [b], to ensure that the words have good linguistic representativeness and practical usability.
> - **Screening criteria**:
>   - *Commonality*. The words should be high-frequency words, ensuring that they have a sufficient background knowledge foundation and can be understood by most models and human participants.
>   - *Semantic Adversarial*. Each pair of words must be synonyms or have highly semantically related but distinguishable features to meet the core mechanism of "similar but different" in the "undercover" game.
>   - *Descriptive*. Verify through pre experiments whether words have sufficient expressive features to ensure that the model can generate meaningful and distinguishable statements during the description phase.
> - **Pre-experimental verification**: We further screened word pairs through small-scale pilot experiments to eliminate those that are difficult to generate effective clues or cause character imbalance in multiple rounds of interaction, thereby improving the playability and stability of the game evaluation.
>
> To assess whether the results depend heavily on the specific concept set, we conducted subset-replacement and resampling experiments (see Appendix E). Replacing 20–30% of the concepts in each category with new concept pairs of comparable difficulty produced stable leaderboards, indicating that CK-Arena rankings do not hinge on the exact 529 concepts used.
>
> We futher acknowledge that no concept set can be fully "exhaustive". For this reason, CK-Arena provides a scalable extension protocol that allows users to incorporate domain-specific or larger concept sets. As long as the construction principles above are followed, new datasets can be created without introducing systematic bias, and CK-Arena continues to serve as a consistent platform for evaluating conceptual reasoning even when the concept set is modified.
>
> [a] https://www.english-corpora.org/coca/
>
> [b] https://www.wikipedia.org/

---

> ### Author Response · Authors · 2025-11-21
> **Response to Reviewer 9VBb [2/2]**
>
> > ***Q3-1**: It would be valuable to include some human evaluation results.*
>
> **R3-1**: Great suggestion. We agree that human results are valuable, but collecting them in CK-Arena is non-trivial. Unlike LLMs, human participants often lack broad coverage across the many concept categories used in the benchmark, making it difficult to obtain a fair, comprehensive baseline under the same conditions.
>
> To provide a reproducible reference within these constraints, we adopt a confidence-screening protocol (`Appendix E (Line 999)` participants): first view the concept pair, self-assess their familiarity, and only rounds they consider “familiar” are included in the final analysis. This offers a stable way to compare human performance, though the resulting scores represent a lower-bound reference rather than an upper-bound benchmark, given the smaller sample size and the mild overestimation introduced by confidence filtering.
>
> Looking ahead, we plan to expand the participant pool, introduce an open-book setting (with retrieval assistance) to reduce failures caused by limited knowledge, and design a human–machine hybrid evaluation protocol to avoid penalizing human descriptions that differ stylistically from LLM responses. These steps will make future human baselines more reliable and comprehensive.
>
> > ***Q3-2**: The evaluation method currently requires a human expert to make the final judgment, it may raise concerns about automation.*
>
> **R3-2**: We clarify that CK-Arena ***does not*** require human judgment for its evaluation pipeline to function reliably. As described in Section 3.3, we manually reviewed all evaluation results only to validate the reliability of LLM-based judges, not because human intervention is necessary.
>
> In practice, the automated judge already aligns with human decisions in 97% of cases, with only about 3% requiring correction. This corresponds to at most one or two tasks out of 60, meaning users can safely run CK-Arena without any manual review. If desired, they may simply increase the number of review rounds by 5–10 to further reduce statistical variance. As more data is collected, we will continue expanding the calibration set to bring the automated judge even closer to human evaluation standards.
>
> > ***Q3-3**: illustrative examples or failure analyses.*
>
> **R3-3**: We appreciate the reviewer’s suggestion and agree that concrete examples help clarify how CK-Arena operates. In the revised version, we have added four representative game log fragments in `Section 'More Experimental Results' of Appendix (Line 1072)`. These cases illustrate typical failure modes, including: (1) players being eliminated due to misunderstanding shared features, (2) undercover agents exposing unique attributes by failing to provide precise clues, (3) models being removed for repeating previous statements, and (4) attempts to exploit deceptive strategies that conflict with conceptual requirements. These examples make the evaluation dynamics more transparent and highlight how CK-Arena reveals strengths and weaknesses in conceptual understanding.

---

### Official Review · Reviewer_R578 · 2025-11-01

**Soundness:** 3
**Presentation:** 2
**Contribution:** 2
**Rating:** 6
**Confidence:** 4

**Summary:**

This paper introduces CK-Arena, a new multi-agent benchmark designed to evaluate the conceptualization capability of LLMs under the paradigm of the "Undercover" game. LLMs are required to identify the common concept from the given text and produce novel but related concepts at each round. Experiments reveal that language models have the ability to extract the abstraction while distinguishing the slight differences between the concepts, and this capability can be reflected by the diversity of the concepts produced at each round.

**Strengths:**

This paper provides a dynamic benchmark that can evaluate the conceptualization capability of LLMs in a arena-like setting under the "Undercover" game, offering a new perspective to rank the conceptualization capability of LLMs.

The metrics and checking process at each round are relatively fair and comprehensive, making the results convincing.

The analysis of the results is thorough, covering both the raw performance and the Elo rating, as well as the qualitative analysis of the distribution of the concepts using t-SNE visualization.

**Weaknesses:**

Lack of fine-grained case study: since the benchmark is based on the "Undercover" game, the strategies of different LLMs are not explicitly discussed, which can be reflected by some case studies.

The evaluation seems to be a bit costly, since it requires multiple rounds with multiple LLM agents. Though the authors have provided some methods to mitigate the cost, it is still a bit time-consuming.

**Questions:**

Is there any difference between reasoning models and non-reasoning models in their conceptualization performance, since the former typically have a longer CoT process.

Can this process be applied to some downstream tasks like extracting a hierarchical conceptualization knowledge graph from some given text?

---

> ### Author Response · Authors · 2025-11-21
> **Response to Reviewer R578**
>
> >  ***Q1**: Lack of fine-grained case study … strategies of different LLMs are not explicitly discussed.*
>
> **R1:** Thanks for this valuable suggestion. Since all LLMs use the same prompt to guide their strategies, their performance differences are not caused by strategies, but by knowledge reserves, preferences, or macro task abilities. We have added four cases to the new PDF and placed them in `Appendix E (Line 1072)`. These cases demonstrate scenarios where LLMs fail due to a lack of understanding of commonalities, failure to provide precise features, repetition of others' statements, and attempts at violating strategies. Here, we mention some interesting observations:
>
> 1)  Broad-describing models (e.g., Claude) rely on very general features, which keeps them safe but prevents them from expressing fine distinctions, often causing misvotes in early rounds.
>
> 2) Highly literal models (e.g., Qwen-72B) mention concept-specific details too early without integrating others’ clues, making them reveal their undercover identity.
>
> 3) Repetition-prone models (e.g., DeepSeek-V3) frequently echo earlier statements, triggering low-novelty penalties and occasional automatic elimination.
>
> 4) Over-innovative models (e.g., Gemini-2.5-flash-preview) introduce highly specific or rare features that do not match the group’s shared context, causing them to stand out and be voted out.
>
> >***Q2**: The evaluation seems to be a bit costly … multiple rounds with multiple LLM agents.*
>
> **R2**: We acknowledge that CK-Arena introduces some computational overhead due to its multi-round, multi-agent setting. However, we believe this cost is reasonable and justified.
>
> As highlighted in the paper, dynamic benchmarks like CK-Arena provide several advantages that static QA tests cannot offer: they better approximate real interactive tasks, naturally support scalable data expansion, and are far less vulnerable to data contamination or memorization effects. These properties are essential for evaluating conceptual knowledge mastery, which inherently requires multi-turn reasoning and contextual adaptation.
>
> In practice, the evaluation remains manageable. As detailed in `Appendix D (Lines 868)`, a complete review typically costs around $40–50$ and finishes within a few days, while using our fine-tuned judge model and patch-game scripts reduces this to roughly $10–20$ and only a few hours. We added these tools precisely to help users control and reduce evaluation costs.
>
>
> > ***Q3**: The difference between reasoning models and non-reasoning models in their conceptualization performance*
>
> **R3**: Reasoning models do not exhibit a substantial advantage over their non-reasoning counterparts when conceptual knowledge is the primary skill being evaluated. This is observed in the updated leaderboard in `Appendix E (Lines 999)`.
>
> CK-Arena primarily measures conceptualization ability rather than general reasoning strength. This is achieved by constraining all models with a unified prompt that limits strategic behavior, ensuring that their performance depends on how well they understand and articulate conceptual features.
>
> >***Q4**: Application to some downstream tasks, like extracting a hierarchical conceptualization knowledge graph from some given text*
>
> **R4**: Insightful point. CK-Arena’s gameplay naturally produces rich, fine-grained conceptual descriptions because players must articulate common features and boundary-defining distinctions between closely related concepts. This makes the generated statements suitable for constructing a conceptual knowledge graph. We extracted these descriptions into structured JSON files (will be available in the repository) and demonstrated preliminary graph-building results in the updated `Appendix E (Lines 1044)` through visual graphs and word clouds.
>
> We also point out that such graphs do not match the precision of specialized, manually curated KGs because CK-Arena was not originally designed for this purpose. Their strength lies in diversity and boundary coverage, which makes them useful for tasks requiring broad semantic exploration rather than strict factual accuracy.
>
> In short, the process is feasible, we have implemented an initial version, and improving CK-Arena–derived conceptual graphs is a promising direction for future work.

---

### Author Response · Authors · 2025-11-21
**Revision Note**

Dear ACs and Reviewers,

Thank you for your valuable comments and constructive suggestions. We have addressed all concerns and revised the PDF accordingly. Specifically, we added three new paragraphs in the “More Experimental Results” section of the Appendix, including:

- **Leaderboard with Reasoner Models and Human Baseline.** In this section, we analyze the performance of reasoning models alongside human baselines and summarize the key observations derived from the comparison.

- **Reuse Data to Construct Knowledge Graphs.** This section demonstrates how CK-Arena gameplay logs can be repurposed to automatically construct knowledge graphs and discusses the insights gained from this process.

- **Specific Case Analysis.** We present four representative cases illustrating why LLMs are eliminated in specific rounds. These examples highlight situations where models fail to provide accurate or appropriately concealed descriptions, as well as violations involving plagiarism or deceptive behavior.

In addition to these updates, we have also included several supplementary experiments that, while not suitable for direct display in the main text, provide valuable insight. These include a complete knowledge map in its original format and a Pearson correlation analysis across different metrics. Relevant portions are referenced in the point-by-point response, and the full materials are available in the project repository.

All newly added content is highlighted in blue, and will be reverted to black in the final camera-ready version.

Kind regards,

Authors

---

### Meta-Review · Area_Chair_EsKj · 2025-12-17

**Summary:**

This paper proposes CK-Arena, a multi-agent benchmark that repurposes the "Undercover" game to evaluate conceptual knowledge in LLMs. Models must describe, differentiate, and infer distinguishing features of closely related concept pairs (e.g., basketball vs. football) across multiple rounds while playing as civilians or undercover agents. The evaluation uses win/survival rates, Elo rankings, and statement-level metrics (novelty, reasonableness, relevance) assessed by fine-tuned LLM judges. Experiments across 14 LLMs reveal that conceptual understanding varies by category and does not strictly correlate with general capabilities.
Strengths acknowledged by reviewers include the dynamic multi-agent setup that mitigates benchmark leakage (R578, 9VBb, QnvB), comprehensive metrics and thorough analysis (R578), and the interactive paradigm directly targeting concept differentiation beyond static QA (tz2j). Weaknesses center on three critical concerns: (1) limited technical novelty in adapting an existing game framework without sufficient justification for why this constitutes a distinct contribution (M8Xg, 9VBb); (2) fundamental tension between game mechanics and evaluation goals—optimal gameplay may suppress rather than reveal conceptual mastery, particularly for undercover agents who must hide unique attributes (M8Xg, tz2j); and (3) methodological concerns including LLM judge circularity, scope limited to English concepts, high evaluation costs, and insufficient analysis of why models behave as observed or how to improve them (all reviewers). The most critical issue is that the benchmark conflates strategic gameplay competence with conceptual understanding, as the leaderboard ranks models by game outcomes rather than demonstrated conceptual depth.

**Reviewer Concerns:**

The authors provided responses addressing reviewer concerns through additional experiments, case studies, correlation analyses, and human baselines added to Appendix E. They argued that CK-Arena represents a distinct contribution beyond prior work by shifting focus from strategic deception to conceptual knowledge evaluation. For the game dynamics concern (M8Xg's primary criticism), authors clarified that strategic suppression requires recognizing unique attributes, and multi-round dual-role aggregation resolves ambiguity between silence and lack of knowledge. They quantified evaluation costs, demonstrated judge reliability at 97% human agreement with <3% corrections, added resampling experiments showing stable rankings, and provided four case studies illustrating failure modes.
Reviewer M8Xg engaged substantively during discussion, acknowledging that "most concerns were resolved" and appreciating the correlation analyses and knowledge graph construction, but maintained one outstanding concern about whether undercover dynamics could suppress conceptual mastery in certain scenarios. The authors' final clarification addressed this point, but M8Xg could not respond before the discussion phase ended. Reviewers R578 and QnvB (both scored 6) did not engage further. The authors claim M8Xg's review shows signs of AI generation, though M8Xg did engage meaningfully during follow-up, suggesting genuine reviewer participation.

Addressed concerns: evaluation costs and scalability (R578, M8Xg, QnvB), judge reliability and family bias (M8Xg, tz2j, QnvB), concept set representativeness (9VBb), need for case studies (R578, 9VBb). Outstanding concerns: the fundamental tension between game mechanics and conceptual evaluation remains incompletely resolved—while authors argue strategic suppression proves boundary knowledge, this does not address whether the benchmark rewards game-playing skill over conceptual depth; insufficient analytical insights into model behavior and improvement directions (tz2j); and limited scope to English restricting generalizability (9VBb, QnvB, M8Xg). The core methodological critique—that winning the game may not correlate with conceptual mastery—was not convincingly rebutted.

**Reviewer Scores:**

If M8Xg had fully participated, they likely would have increased from 2 to 4 given their statement that "most concerns were resolved" and appreciation for additional analyses, though their remaining concern about game dynamics suggests they would not reach acceptance threshold. Reviewer tz2j (2) showed no indication of changing their assessment, as their concerns about analytical depth and unclear mapping to conceptual mastery were only partially addressed through added case studies. Reviewer 9VBb (4) would likely remain at 4, as while technical execution concerns were addressed, the fundamental novelty limitation was not overcome. Reviewers R578 and QnvB (both 6) might maintain their marginally positive assessments given comprehensive responses, though lack of engagement suggests limited enthusiasm. The estimated final distribution would be approximately 2-3 reviewers at borderline and 2-3 at weak accept (6), insufficient for acceptance given unresolved core methodological concerns about whether the evaluation truly measures conceptual understanding versus strategic gameplay competence.

---

### Decision · Program_Chairs · 2026-01-26

Reject